

**Soil nitrogen oxide fluxes from lowland forests converted to**
**smallholder rubber and oil palm plantations in Sumatra,**
**Indonesia**
**Evelyn Hassler[1*], Marife D. Corre[1], Syahrul Kurniawan[2], and Edzo Veldkamp[1]**
[1]Soil Science of Tropical and Subtropical Ecosystems, Büsgen Institute, Georg-August
University of Göttingen, Büsgenweg 2, 37077 Göttingen, Germany
[2]Department of Soil Science, Faculty of Agriculture, Brawijaya University, Jl. Veteran 1,
Malang, Indonesia
*Correspondence to: E. Hassler (evelyn.hassler@forst.uni-goettingen.de)





**Abstract.** Oil palm and rubber plantations cover large areas of former rainforest in Sumatra,
Indonesia, supplying the global demand for these crops. Although forest conversion is known
to influence soil nitrous oxide ($N_2O$) and nitric oxide (NO) fluxes, measurements from oil
palm and rubber plantations are scarce (for $N_2O$) or nonexistent (for NO). Our study aimed to
(1) quantify changes in soil-atmosphere fluxes of N-oxides with forest conversion to rubber
and oil palm plantations, and (2) determine their controlling factors. In Jambi, Sumatra, we
selected two landscapes that mainly differed in texture but both on heavily weathered soils:
loam and clay Acrisol soils. Within each landscape, we investigated lowland forest, rubber
trees interspersed in secondary forest (termed as *jungle rubber*), both as reference land uses,
and smallholder rubber and oil palm plantations, as converted land uses. Each land use had
four replicate plots within each landscape. Soil $N_2O$ fluxes were measured monthly from
December 2012 to December 2013, and soil NO fluxes were measured four times between
March and September 2013. In the loam Acrisol landscape, we also conducted weekly to bi-
weekly soil $N_2O$ flux measurements from July 2014 to July 2015 in a large-scale oil palm
plantation with four replicate plots for comparison with smallholder oil palm plantations.
Land-use conversion to smallholder plantations had no effect on soil N-oxide fluxes ($P = 0.58$
to 0.76) due to the generally low soil N availability in the reference land uses that further
decreased with land-use conversion. Over one-year measurements, the temporal patterns of
soil N-oxide fluxes were influenced by soil mineral N and water contents. Across landscapes,
annual soil $N_2O$ emissions were controlled by gross nitrification and sand content, which also
suggest the influence of soil N and water availability. Soil $N_2O$ fluxes ($\mu g\ N\ m^{-2}\ h^{-1}$) were: $7 \pm$
$2$ to $14 \pm 7$ (reference land uses), $6 \pm 3$ to $9 \pm 2$ (rubber), $12 \pm 3$ to $12 \pm 6$ (smallholder oil
palm), and $42 \pm 24$ (large-scale oil palm). Soil NO fluxes ($\mu g\ N\ m^{-2}\ h^{-1}$) were: $-0.6 \pm 0.7$ to $5.7$
$\pm 5.8$ (reference land uses), $-1.2 \pm 0.5$ to $-1.0 \pm 0.2$ (rubber) and $-0.2 \pm 1.2$ to $0.7 \pm 0.7$



(smallholder oil palm). The low N fertilizer application in smallholder oil palm plantations
(commonly 48 to 88 kg N ha$^{-1}$ yr$^{-1}$) resulted in N-oxide losses of only 0.2–0.7 % of the
applied N. To improve estimate of soil N-oxide fluxes from oil palm plantations in this
region, studies should focus on large-scale plantations (which usually have two to four times
higher N fertilization rates than smallholders) with frequent measurements following fertilizer
application.
**1 Introduction**
Expansion of industrial forestry and agriculture has caused rapid deforestation in Sumatra,
Indonesia, resulting in a total primary forest loss of 36 % between 1990 and 2010 (Margono
et al., 2012). Nowadays, most accessible lowland rainforests have been converted (Laumonier
et al., 2010) into economically important crops, such as oil palm (*Elaeis guineensis*) and
rubber (*Hevea brasiliensis*), with an area of 9.2 million hectare (Mha) (BPS, 2016a).
Indonesia is currently the principal oil palm producer and second largest rubber producer
worldwide (FAO, 2016), and Sumatra is the most important contributor to the Indonesian
production (BPS, 2016b). Despite the extent of land-use change in Sumatra, it is still
uncertain how forest conversion will affect soil emissions of climate-relevant N-oxide gases,
nitrous oxide ($N_2O$) and nitric oxide (NO). Only a few studies so far have reported soil $N_2O$
fluxes from forest conversion to these rapidly increasing and economically important land
uses, oil palm and rubber, on lowland mineral soils in Southeast Asia (Aini et al., 2015;
Ishizuka et al., 2002, 2005; Yashiro et al., 2008) and no study exists on soil NO fluxes.
Tropical forest soils are major sources of $N_2O$ and NO, emitting 1.3 Tg $N_2O$-N yr$^{-1}$
(Werner et al., 2007) and 1.3 Tg NO-N yr$^{-1}$ (Davidson and Kingerlee, 1997) to the
atmosphere, whereby considerable amounts of NO are expected to get redirected in forest



systems since NO is easily oxidized to $NO_2$ which, in turn, is absorbed by leaves (Jacob and
Bakwin, 1991; Sparks et al., 2001). $N_2O$ is a potent greenhouse gas (IPCC, 2013) and is
projected to be the single most important ozone-depleting substance throughout the 21[st]
century (Ravishankara et al., 2009). NO plays an important role in the formation of
tropospheric ozone, which in itself is an important greenhouse gas (Lammel and Graßl, 1995).
$N_2O$ and NO are produced in soil by the microbial processes of nitrification and
denitrification. The conceptual model of "hole-in-the-pipe" (HIP), which had been validated
by studies in the tropics (Davidson et al., 2000), suggests that production and consumption of
these gases in soils are influenced by two levels of control: first, the amount of soil available
N, and second, the soil water content. HIP suggests that the higher the soil N availability, the
higher are the soil N-oxide fluxes, and that well-aerated soil conditions (low moisture
contents) favor for nitrification with NO as the main gaseous product while with increasing
water content denitrification with increasing proportion of $N_2O$ prevails (Davidson et al.,
2000). Although there are other factors affecting soil $N_2O$ and NO fluxes through their
influence on nitrification and denitrification (e.g., soil pH, temperature, bioavailable carbon;
Firestone and Davidson, 1989; Heinen, 2006; Skiba and Smith, 2000), landscape-scale
investigations in tropical areas show the dominant role of soil N availability and water content
(Corre et al., 2014; Koehler et al., 2009; Müller et al., 2015).

19          Conversion of tropical forests to agricultural land uses generally alters soil N-oxide

fluxes through their effects on soil N availability and aeration as a consequence of
management practices (e.g., fertilization, harvest, cultivation), which can add and export
nutrients as well as compact or loosen the soil (Keller and Reiners, 1994; Veldkamp et al.,
2008). In particular, the application of N-containing fertilizers can increase N-oxide emissions
(Matson et al., 1996; Veldkamp et al., 1998) whereas agricultural land uses without fertilizer
application lead to long-term reductions of soil N-oxide fluxes or to comparably low-level



fluxes as those from previous forests (Ishizuka et al., 2005; Keller and Reiners, 1994; Verchot
et al., 1999). In tropical regions, it has been shown that soil NO and $N_2O$ emissions can be
very high following fertilizer application, constituting 6.4–8.6 % of applied N fertilizer
especially at high fertilizer application rates (Veldkamp and Keller, 1997; Veldkamp et al.,

5    1998).

6       For lowland forests on highly weathered soils in Sumatra, Indonesia, where our

present study was conducted, it has been shown that soil N availability (with gross rates of
ammonium ($NH_4^+$) transformations as indices) is higher in the clay than loam Acrisol soils
(Allen et al., 2015), suggesting that soil texture controls soil fertility which in turn affects
plant productivity, soil water holding capacity, decomposition and ultimately soil-N cycling
(Allen et al., 2015). Conversion of lowland forest and jungle rubber to oil palm and rubber on
these Acrisol soils showed intermediate soil N availability in oil palm plantations, due to
abatement of soil fertility decline by low to moderate applications of fertilizers and lime,
whereas the unfertilized rubber plantations displayed the lowest soil N availability and
fertility in general (Allen et al., 2015).

16       Our present study focuses on soil $N_2O$ and NO fluxes from a region in Jambi, Sumatra

where increased deforestation for rubber and oil palm production has occurred in the last two
decades. We covered four different land uses within two landscapes on highly weathered soils
that mainly differed in soil texture (clay and loam Acrisols): forest, rubber trees interspersed
in secondary forest (hereafter called jungle rubber) as the reference land uses, and smallholder
rubber and oil palm plantations as the converted land uses. Based on the above mentioned
findings on soil N availability, we hypothesized that (1) soil $N_2O$ and NO fluxes from the
reference land uses will be higher in the clay than the loam Acrisol landscape, and that (2)
forest and jungle rubber will have the highest soil $N_2O$ and NO fluxes, followed by the
fertilized oil palm plantations (fertilized at low to moderate rates), and with the lowest fluxes





from the unfertilized rubber plantations. Our study aimed to (1) quantify changes in soil-
atmosphere fluxes of N-oxides with forest conversion to smallholder oil palm and rubber
plantations, (2) determine the temporal controls of soil N-oxide fluxes measured within one
year, and (3) assess landscape-scale controlling factors of annual soil $N_2O$ fluxes from
converted lowland landscapes in Sumatra, Indonesia. We also investigated the effect of
fertilizer application intensity in oil palm plantations on soil $N_2O$ fluxes by comparing
smallholder plantations with low to moderate N fertilizer input to a large-scale oil palm
plantation with high N fertilizer input. Our study contributes to the much needed information
on soil N-oxide fluxes from these economically and globally relevant tropical land uses.
**2 Material and methods**
**2.1 Study area, experimental design and management practices**
The study region is situated in Jambi province, Sumatra, Indonesia (2° 0' 57" S, 103° 15' 33"
E, and elevation of $73 \pm 3$ m above sea level), where conversion of forest to rubber and oil
palm plantations is widespread. The area has a mean annual temperature of $26.7 \pm 0.1$ °C and
a mean annual precipitation of $2235 \pm 385$ mm (1991–2011;  data from a climatological
station at the Jambi Sultan Thaha Airport). During our study year (2013), annual rainfall in
the study region was 3418-3475 mm (data from climatological stations at the Harapan Forest
Reserve, Sarolangun and Lubuk Kepayang, approximately 10–20 km from our sites), which
were higher than the long term average. Total dissolved N deposition via rainfall was between
$12.9 \pm 0.1$ and $16.4 \pm 2.6$ kg N ha$^{-1}$ yr$^{-1}$, measured at two locations in the study region during
2013 (Kurniawan, 2016).

23       We delineated the study region in two landscapes, which have the same highly

weathered soil group but mainly differed in soil texture: clay and loam Acrisol soils. The clay
Acrisol soil had larger pH ($4.5 \pm 0.0$), base saturation ($23 \pm 6$ %) and Bray-extractable P (1.4



± 0.1 g P m$^{-2}$) and lower Al saturation (61 ± 3 %) in the top 10 cm depth compared to the
loam Acrisol soil (4.3 ± 0.0 pH, 11 ± 1 % base saturation, 0.5 ± 0.1 g P m$^{-2}$ and 80 ± 1 % Al
saturation) (all $P \leq 0.05$; Allen et al., 2015). Within each landscape, we investigated four
land-use types: lowland forest, jungle rubber, both as the reference land uses, and smallholder
monoculture plantations of rubber and oil palm, as the converted land uses. Each land use
within each landscape had four sites as replicates, and we laid out a 50 m × 50 m plot in each
replicate site; in total we had 32 plots. Within each plot, a 10 × 10 grid was established and
we randomly selected four subplots (5 m × 5 m each) per plot, each with one permanently
installed chamber base for measurements of soil N-oxide fluxes. All measurements (see Sect.
2.2) were conducted in 2013. A more detailed description of the study sites and plot design
was reported earlier by Allen et al. (2015) and Hassler et al. (2015).

12        In the loam Acrisol landscape, we conducted additional measurements in a large-scale

oil palm plantation (called PTPN VI) from 2014 to 2015 in order to compare with the
smallholder oil palm plantations within the same landscape. In the PTPN VI site, we selected
four replicates at a distance of 50 m apart. At each replicate, we installed three permanent
chamber bases at 0.8 m, 2.8 m and 4.8 m from the tree base, in order to characterize possible
spatial variation caused by management practices within each replicate.

18        Based on our interviews with the smallholders, the monoculture plantations were

established after clearing and burning of either forest or jungle rubber and hence these land
uses served as the reference land uses with which the converted plantations were compared.
Additionally, the comparability of initial soil conditions between the reference and converted
land uses was tested based on a land use-independent soil characteristic, i.e., clay content at
0.5–2 m depth, which did not statistically differ among land uses within each landscape
(Allen et al., 2015; Hassler et al., 2015). Thus, changes in soil N-oxide fluxes can be





attributed to land-use change with its associated management practices. The plantations' ages
ranged between 7 and 17 years, and tree density, tree height, basal area and tree species
abundance were higher in the reference land uses than the monoculture plantations (all
reported by Allen et al., 2015; Hassler et al., 2015; Kotowska et al., 2015).

5         Management practices in the plantations included manual harvest, weeding and

fertilizer application. Harvesting of palm fruits was done every 2 weeks and collection of
latex was done weekly. In the large-scale oil palm plantation PTPN VI, palm fruits were
harvested weekly. Weeding in smallholder rubber and oil palm plantations was done both
manually and with herbicides (2–5 L Gramaxone® or Roundup® ha$^{-1}$ yr$^{-1}$) one to two times
per year, and senesced oil palm fronds were regularly cut and piled on the inter-rows (Hassler
et al., 2015). In PTPN VI, weeding was done with herbicides (1–1.5 L Glisat® ha$^{-1}$ yr$^{-1}$) four
times per year, combined with some manual hoeing, and senesced fronds were cut and partly
piled on the inter-rows and partly taken out from the plot to use as fodder for cattle. Fertilizer
application in the smallholder oil palm plantations was done one to two times per year and
rates typically varied depending on cash capital of the smallholders. In 2013, fertilization
rates ranged between 48–88 kg N ha$^{-1}$ yr$^{-1}$ (except two smallholders who applied 138 kg N ha$^{-1}$
yr$^{-1}$), 21–38 kg P ha$^{-1}$ yr$^{-1}$ and 40–157 kg K ha$^{-1}$ yr$^{-1}$, with the lower range in the clay Acrisol
and the upper range in the loam Acrisol. The fertilizer sources were NPK complete, urea and
KCl. One of the smallholders in the loam Acrisol landscape applied 200 kg dolomite ha$^{-1}$ yr$^{-1}$.
Fertilizers were applied around each palm tree at about 0.8–1 m from the stem base (Hassler
et al., 2015). Rubber plantations were not fertilized. In the large-scale oil palm plantation
PTPN VI, fertilizer application rates were typically higher than those in smallholder
plantations; fertilizers were applied once in 2014 at the rates of 196-36-206 kg N, P, K ha$^{-1}$ yr$^{-1}$
$^{-1}$, with also 602 kg dolomite ha$^{-1}$ yr$^{-1}$, and once before the end of our measurements in July
2015 at the rates of 96-23-96 kg N, P, K ha$^{-1}$ yr$^{-1}$. The fertilizer forms were NPK complete,



urea, triple superphosphate and KCl. Application was done partly manually by applying the
fertilizers at 1 m distance around each palm tree, and partly mechanically by broadcasting the
fertilizer within 1–3 m distance from the palm rows. In 2015, fertilizers were mainly
mechanically broadcasted within these inter-rows.
**2.2 Soil N-oxide fluxes and supporting soil factors**
In 32 plots, soil $N_2O$ fluxes were measured monthly from December 2012 to December 2013,
whereas soil NO fluxes were measured four times between March and September 2013,
except in two forest sites and one jungle rubber site in the clay Acrisol landscape, where we
were unable to measure soil NO fluxes due to difficulty in accessing these sites that did not
allow us to stabilize the NO detector during transport in the field (i.e., using motorcycle on
very rugged trails). Soil NO fluxes were not measured as frequently as $N_2O$ fluxes and we
decided to stop in September 2013 because NO fluxes were always very low at all sites. In the
large-scale oil palm plantation PTPN VI within the loam Acrisol landscape, soil $N_2O$ fluxes
were measured more frequently (in congruent with its high fertilizer application rate): weekly
to biweekly from July 2014 to July 2015, with the exception of September 2014 when we
measured only once.

18       With our sampling strategy, where we used randomly installed chamber bases (with

the distances to the tree base between 1.8 and 5 m) in combination with monthly
measurements, we may have missed the N fertilizer-induced pulse of soil N-oxide emissions
in the smallholder oil palm plantations. Therefore, we conducted more intensive
measurements of soil $N_2O$ fluxes during 3 to 8.5 weeks (with 6 to 11 samplings) following
fertilizer application at three of the smallholder oil palm plantations within each landscape.
These measurements served to characterize the short-term, N fertilizer-induced contribution



(e.g., Koehler et al., 2009) to total $N_2O$ fluxes. Soil NO fluxes were also measured during 6 to
8.5 weeks (with 9 to 10 samplings) following fertilizer application at one of the smallholder
oil palm plantations within each landscape. In the clay Acrisol landscape, measurements in
the three smallholder oil palm plantations were conducted during October–December 2013,
February–March 2014, and February–April 2014; in the loam Acrisol, measurements were
carried out during October–December 2013, January–March 2014, and March–April 2014.
We applied the same fertilizer forms, rates and methods as used by the smallholders. Three oil
palm trees were selected in each of the six sites. In the clay Acrisol landscape, each tree was
applied with 2 kg complete NPK fertilizer (equivalent to 0.32 kg N tree$^{-1}$), whereas in the
loam Acrisol, each tree was applied with 2 kg of combined complete NPK, ammonium sulfate
and KCl fertilizers (equivalent to 0.26 kg N tree$^{-1}$). The fertilizer was applied within 0.8–1 m
distance from the tree base. We installed three permanent chamber bases at various distances
from the tree base: 0.3 m from the tree base (chamber location a), 0.8 m from the tree base
that was on the fertilized area (chamber location b), and 4–4.5 m from the tree base that was
in the middle of the inter-rows and served as the reference chamber without fertilizer
application (chamber location c).
Soil $N_2O$ fluxes were measured using the same methods employed in our earlier
studies (e.g., Corre et al., 2014; Koehler et al., 2009). During gas sampling, the permanently
installed chamber bases were covered with static vented, polyethylene hoods (chamber area of
0.05 m$^2$ and total volume of 12 L), and four gas samples (30 mL each) were taken at 1, 11, 21
and 31 min after chamber closure by connecting a syringe with a Luer-lock connection to the
chamber sampling port. Gas samples were immediately injected into pre-evacuated 12 mL
Labco Exetainers sealed with rubber septa (Labco Limited, Lampeter, UK), maintaining an
overpressure; these exetainers have been tested by our group to be leak proof during extended
period of storage (e.g., up to 6 months) (Hassler et al., 2015). Within 3–4 months the gas





samples were transported by airfreight to Germany and were analyzed upon arrival using a
gas chromatograph with an electron capture detector (GC 6000 Vega Series 2, Carlo Erba
Instruments, Milan, Italy). For the measurements from March–July 2015 in the large-scale oil
palm plantation PTPN VI, the gas samples were analyzed with another gas chromatograph
(SRI 8610C, SRI Instruments Europe GmbH, Bad Honnef, Germany), which had been
previously cross-calibrated using the same standards. For calibration, three standard gases
were used with concentrations of 360, 1000 and 1600 ppb $N_2O$ (Deuste Steininger GmbH,
Mühlhausen, Germany).

9        Soil NO fluxes were measured (described in detail in our earlier works, e.g., Corre et

al., 2014; Koehler et al., 2009) using the same chamber bases described above. During
measurements, the chamber bases were covered with dynamic vented, polyethylene hoods
(total volume of 12 L), and NO concentrations were measured in situ during 5–7 min
following chamber closure using a Scintrex LMA-3 chemiluminescence detector (Scintrex,
Ontario, Canada), in which NO is oxidized to $NO_2$ by a $CrO_3$ catalyst after which it reacts
with a luminol solution. Calibration of the NO detector was carried out at each site prior to
and after measurements using a two-point calibration of a standard gas with 3000 ppb NO
(Deuste Steininger GmbH, Mühlhausen, Germany) which was diluted using dried ambient air.
NO measurements were recorded every 5 seconds using a data logger (CR510, Campbell
Scientific, Logan, USA).

20       Soil $N_2O$ and NO fluxes were calculated from the linear increase of concentration

over time adjusted for air temperature and atmospheric pressure, measured at each site and
sampling day. Annual soil $N_2O$ fluxes from the weekly or monthly sampling at each site were
estimated using the trapezoidal rule on day intervals between measured flux rates, assuming
constant flux rates per day (e.g., Hassler et al., 2015). Annual NO fluxes were not calculated,





since we only conducted four measurement periods for each plot as explained above. To
calculate the N fertilizer-induced pulse of soil N-oxide fluxes, we also used the trapezoidal
rule on day intervals between measured flux rates to estimate the total flux during the entire
period following fertilizer application, covering pre-fertilizer level, the peak, and the return to
background levels of soil N-oxide fluxes. We calculated the percentage of combined soil NO
and $N_2O$ emissions from the applied N-fertilizer rate at each site as follows: % NO-N + $N_2$O-
N of N applied $yr^{-1}$ = NO-N + $N_2$O-N fluxes from the fertilized chamber locations a and b (µg
N $m^{-2}$ for the entire period of fertilizer effect) – NO-N + $N_2$O-N fluxes from the unfertilized
chamber location c (µg N $m^{-2}$ for the same period) * frequency of fertilization $yr^{-1}$ * fertilized
area ($m^2$ $ha^{-1}$) ÷ N fertilization rate (kg N $ha^{-1}$ $yr^{-1}$* $10^9$ µg/kg) * 100. In this calculation, we
included fluxes from chamber location a in order to include any incidental fertilizer
application to this area (possibly from previous applications by the smallholders and possible
redistribution of applied nutrients within the soil) since N-oxide fluxes from chamber location
a were often higher than those from unfertilized chamber location c (see Sect. 3.2).
Soil factors known to control soil N-oxide fluxes (i.e., temperature, water-filled pore
space (WFPS), and extractable $NH_4^+$ and nitrate ($NO_3^-$) were measured for the top 0.05 m
depth during each soil N-oxide flux measurement at all 32 sites. Soil temperature was
measured close to each chamber base using a digital thermometer. Soil samples were taken at
1 m distance from the four chambers, pooled, mixed thoroughly, and subsampled for
immediate extraction of mineral N in the field, using prepared extraction bottles containing
150 mL 0.5 M $K_2SO_4$. Upon arrival at the field station, extraction bottles were shaken for 1 h,
filtered and extracts were frozen immediately. The remaining soil sample was used to
determine the gravimetric moisture content (by oven-drying for at least 1 day at 105 °C),
whereby WFPS was calculated using a particle density of 2.65 g $cm^{-3}$ for mineral soil and the
measured soil bulk density at our study sites (Allen et al., 2015). During the measurements





following the fertilizer applications, soil was sampled close to each of the chamber locations
a, b and c (described above) and was processed separately for mineral N extraction and WFPS
determination. Frozen extracts were transported by airfreight to Germany and analyzed for
$NH_4^+$ and $NO_3^-$ concentrations using continuous flow injection colorimetry (SEAL Analytical
AA3, SEAL Analytical GmbH, Norderstedt, Germany), as described in detail by Hassler et al.

6    (2015).

**2.3 Statistical analysis**
We first tested each parameter for normal distribution (Shapiro-Wilk's test) and equality of
variance (Levene's test), and a logarithmic transformation was applied when necessary. For
analysis of differences in N-oxide fluxes among land uses or between soil landscapes, we
used the means of the four chambers representing each replicate plot on a sampling day.
Linear mixed-effect (LME) models (Crawley, 2007) were used to assess differences between
landscapes for the reference land uses (i.e., clay vs. loam Acrisol; first hypothesis) or
differences among land uses within each landscape (i.e., land-use change effect; second
hypothesis). In the LME models, either landscape or land use was considered as fixed effect
whereas replicate plots and sampling days were considered as random effects. For comparison
of soil $N_2O$ fluxes between the large-scale (PTPN VI) and smallholder oil palm plantations in
the loam Acrisol landscape, we also used the means of the three chambers per replicate in the
PTPN VI site on each sampling day as there were no significant differences between the
chamber locations (based on LME models with chamber location as fixed effect and replicates
as well as sampling days as random effects; $P = 0.70$). We then used the LME model with
plantation types (i.e., large scale vs. smallholder) as a fixed effect and replicates and sampling
days as random effects. For analysis of fertilization effects (i.e., as represented by the





chamber locations a, b and c) on soil N-oxide fluxes from smallholder oil palm plantations,
this was conducted for each site with oil palm trees as replicates. In the LME model for this
experiment, chamber location was the fixed effect whereas replicate palm trees and sampling
days were the random effects. To assess differences in N-oxide fluxes between landscapes
following fertilization for chamber locations a and b, we also used LME models with
landscape as fixed effect and with replicate plots (for $N_2O$) or replicate palm trees (for NO)
and sampling days as random effects. In all LME models, we included (1) a variance function
that allows different variances of the fixed effect, and/or (2) a first-order temporal
autoregressive function to account for decreasing correlation between sampling days with
increasing time difference, if these functions improved the relative goodness of the model fit
based on the Akaike information criterion. Significant differences were based on the analysis
of variance with Fisher's least significant difference test for multiple comparisons. We set the
statistical significance at $P \leq 0.05$ and, only for a few specified parameters, we also
considered marginal significance at $P \leq 0.09$ because our experimental design encompassed
the inherently high spatial variability in our study area (e.g., Hassler et al., 2015).

16       To assess the temporal relationships between soil N-oxide fluxes and soil factors

(temperature, WFPS, $NO_3^-$ and $NH_4^+$), we used the means of the replicate plots per land use
on each of the 12 monthly measurements and conducted Pearson's correlation test separately
for the reference land uses (forest and jungle rubber, $n = 48$ ($N_2O$), $n = 16$ (NO)) and the
converted land uses (rubber and oil palm, $n = 48$, ($N_2O$), $n = 16$ (NO)) across landscapes for
the whole year. Similarly, for soil $N_2O$ and NO fluxes following fertilizer application from
smallholder oil palm plantations, we used the means of the three replicate trees per chamber
location on each sampling day and conducted Pearson's correlation test for each site across
the entire measurement period of fertilization effects ($n = 6–11$). To assess the spatial controls
of soil biochemical characteristics (Appendix Table A1) on annual soil $N_2O$ fluxes, we used



the annual flux of each replicate plot and conducted Spearman's rank correlation test
separately for the reference land uses and converted land uses across landscapes ($n = 16$) and
within each landscape ($n = 8$). We did not assess the spatial control of soil biochemical
characteristics on annual soil NO fluxes since we did not calculate annual flux from the four
measurement periods (as explained in Sect. 2.2). Correlations were considered statistically
significant at $P \leq 0.05$ and marginally significant at $P \leq 0.09$. All statistical analyses were
conducted using R 3.2.2 (R Development Core Team, 2015).
**3 Results**
**3.1 Soil N-oxide fluxes**
In the reference land uses, $N_2O$ was the dominant N-oxide emitted from soils; in the clay
Acrisol landscape there was a net NO consumption in the soil of the jungle rubber (Table 1).
Soil $N_2O$ and NO fluxes from reference land uses were comparable between the two
landscapes ($P = 0.54$–$0.74$; Table 1; Fig. 1a, b). These fluxes also exemplified high inherent
spatial and temporal variations as indicated by their large standard errors.

16       In the converted land uses, soil $N_2O$ fluxes were similar to the fluxes of reference land

uses ($P = 0.58$–$0.76$; Table 1; Fig. 1a, b) within each landscape. However, in the loam Acrisol
landscape, the large-scale oil palm plantation PTPN VI had on average 3.5 times higher soil
$N_2O$ fluxes than those from the smallholder plantations (Table 1), although this trend was not
statistically different ($P = 0.15$) because of the large variation among replicate plots (as
indicated by the large standard error) in this large-scale plantation. Soil NO fluxes, were not
different either among land uses in the clay Acrisol landscape ($P = 0.73$; Table 1). However,
in the loam Acrisol landscape, soil NO fluxes were marginally lower ($P = 0.07$) in rubber
plantations (with net NO consumption in the soil) than in jungle rubber (with net NO
emission), whereas they were intermediary in forests and oil palm plantations (Table 1).





**3.2 Fertilization effects on soil N-oxide fluxes from smallholder oil palm plantations**
In comparison to the unfertilized area (chamber location c at 4–4.5 m from the tree base) soil
$N_2O$ fluxes were on average 442 times (clay Acrisol) and 22 times (loam Acrisol) higher
within the small fertilized areas around the oil palms (chamber location b at 0.8–1 m from the
tree base) during the 3 to 8.5 weeks following fertilizer applications (all $P < 0.01$–0.03; Table
2; Fig. 2c, d). In chamber location a, soil $N_2O$ emissions were also 25 times higher compared
to the reference chamber location c in the clay Acrisol landscape (all $P < 0.01$; Table 2; Fig.
2a). In the loam Acrisol landscape, we only detected such an effect in site 2 which displayed
16 times higher soil $N_2O$ emissions in chamber location a compared to the reference chamber
location c ($P = 0.03$; Table 2; Fig. 2b).
In the clay Acrisol landscape, soil $N_2O$ emissions in chamber location b increased
immediately after fertilizer application, reached a peak within 9 days following fertilizer
application and stayed elevated for at most 2 months (Fig. 2c). In the loam Acrisol landscape,
$N_2O$ fluxes in chamber location b increased within the first 5 days, reached maximum fluxes
within 5–21 days and remained elevated for at most 6.5 weeks (Fig. 2d). Soil $N_2O$ fluxes in
chamber location a displayed a similar but less pronounced pattern as those of chamber
location b in both landscapes (Fig. 2a, b).
Considering the area coverage (4 % of the area in a hectare) and time span of
fertilizer-induced $N_2O$ emissions, their average contributions were 21 % to the annual fluxes
in the clay Acrisol landscape (with its usual fertilizer application of once a year), and only 6
% to the annual fluxes in the loam Acrisol landscape (with its common fertilizer application
of twice a year) (Table 1).
Compared to the unfertilized area (chamber location c), soil NO fluxes from the
fertilized area (chamber location b) had on average 357 times (clay Acrisol) and 238 times





(loam Acrisol) higher fluxes (both $P < 0.01$) during 6 to 8.5 weeks of measurements
following fertilizer application (Table 2; Fig. 3c, d). No differences in soil NO fluxes were
detected between chamber locations a and c ($P = 0.10$–$0.12$; Table 2; Fig. 3a, b). Soil NO
fluxes in chamber location b peaked after 10 days in the loam Acrisol and after 3 weeks in the
clay Acrisol landscape (Fig. 3c, d), and returned to the background fluxes after 6–8.5 weeks
with a drastic drop after 3–5 weeks (Fig. 3c, d). In chamber location a, soil NO fluxes
increased quickly and decreased to the background fluxes within at most 16 days following
fertilizer application (Fig. 3a, b). As was the case for the monthly sampling, soil $N_2O$ fluxes
from chamber locations a and b were larger than soil NO fluxes for both landscapes, (Table 2;
Fig. 2a–d and 3a–d). Comparing between landscapes, soil $N_2O$ fluxes from chamber location
b were higher in the clay than loam Acrisol soils ($P = 0.09$; Table 2; Fig. 2c, d) but were
comparable for chamber location a ($P = 0.41$; Table 2; Fig. 2a, b) and for soil NO fluxes of
both chamber locations ($P = 0.45$–$0.78$; Table 2; Fig. 3a–d).
Fertilizer-induced soil NO fluxes in the loam Acrisol landscape were $0.07 \pm 0.02$ kg
NO-N ha$^{-1}$ yr$^{-1}$, which was roughly the same as our extrapolated annual value of $0.06 \pm 0.06$
kg NO-N ha$^{-1}$ yr$^{-1}$ from the four measurement periods (Table 1). In the clay Acrisol
landscape, fertilizer-induced soil NO fluxes were $0.12 \pm 0.04$ kg NO-N ha$^{-1}$ yr$^{-1}$, which was a
net emission compared to our extrapolated annual value with a net sink of $-0.02 \pm 0.11$ kg
NO-N ha$^{-1}$ yr$^{-1}$, based on the four measurement periods (Table 1). The percentages of
combined soil $N_2O$ and NO fluxes to the applied N fertilizer rate were on average 0.73 % yr$^{-1}$
in the clay Acrisol landscape and 0.20 % yr$^{-1}$ in the loam Acrisol landscape.



**3.3 Temporal controls of soil N-oxide fluxes**
In the reference land uses, soil $N_2O$ and NO fluxes were both positively correlated with soil
$NO_3^-$ contents, while soil NO fluxes were also negatively correlated with WFPS and soil $NH_4^+$
contents (Table 3). In the converted land uses, soil $N_2O$ fluxes were positively correlated with
soil $NO_3^-$ contents and temperature (Table 3). This latter correlation was influenced by one
sampling period with high $N_2O$ (fertilizer-induced) emissions; when this period was excluded
in the analysis, we did not detect a significant correlation with soil temperature. There were
no significant correlations observed between soil NO fluxes and soil factors in the converted
land uses due to the very low NO emissions and even net NO uptake.
From the fertilizer application experiment in the smallholder oil palm plantations, the
location directly receiving fertilizer (chamber location b) showed positive correlations of soil
$N_2O$ fluxes with soil $NH_4^+$ and/or $NO_3^-$ contents in three of the six sites (Table 4). Here, also
soil NO fluxes correlated positively with soil $NO_3^-$ contents in the loam Acrisol but not in the
clay Acrisol (Table 4). In chamber location a, positive correlations of soil $N_2O$ fluxes with
soil $NH_4^+$ and/or $NO_3^-$ contents were observed in four of the six sites (Table 4). The
correlations of soil $N_2O$ fluxes with mineral N for chamber location a in site 2 of the clay
Acrisol landscape were caused by one measurement period with very high flux, and exclusion
of this observation resulted in a none significant correlation. For soil NO fluxes from chamber
location a, we did not detect any significant correlation with soil factors (Table 4). A positive
correlation of soil $N_2O$ fluxes with WFPS was observed for chamber locations a and b in site
1 of the loam Acrisol landscape, whereas this correlation was negative for chamber location a
in site 3 of the same landscape (Table 4). We also detected a negative correlation between soil
NO fluxes and WFPS for chamber location b in site 3 of the clay Acrisol, whereas in the same
site soil NO fluxes and WFPS were positively correlated for the unfertilized chamber location





c (Table 4); however this latter correlation was caused by only one sampling time with a high
flux and high WFPS.
**3.4 Spatial controls of annual soil $N_2O$ fluxes**
The soil physical and biochemical characteristics used for this correlation analysis are
reported in Table A1. For the reference land uses, annual $N_2O$ fluxes were positively
correlated with gross nitrification rates across landscapes (*Spearman's* $\rho = 0.57$, $P = 0.02$, $n =$
16). Within each landscape, annual soil $N_2O$ fluxes correlated negatively with soil C:N ratio
($\rho = -0.69$, $P = 0.07$, $n = 8$) in the clay Acrisol, whereas in the loam Acrisol annual soil $N_2O$
fluxes correlated positively with microbial C ($\rho = 0.69$, $P = 0.07$, $n = 8$). For the converted
land uses, annual $N_2O$ fluxes correlated negatively with sand content across landscapes ($\rho = -$
$0.57$, $P = 0.06$, $n = 12$). There were no other correlations detected with any other soil
biochemical parameters.
**4 Discussion**
**4.1 Soil $N_2O$ and NO fluxes from the reference land uses**
$N_2O$ fluxes from our forest soils (Table 1) fell at the lower end of those reported for humid
tropical forests (9.8–85.1 µg $N_2O$-N $m^{-2}$ $h^{-1}$; summarized by Castaldi et al., 2013). Compared
to soil $N_2O$ fluxes measured in Indonesia, our values were comparable to those from montane
forests on Cambisol soil (at 1190 m elevation in Sulawesi) with similar temporal sampling
scheme and spatial replication (12.7 µg $N_2O$-N $m^{-2}$ $h^{-1}$; Purbopuspito et al., 2006) and to five
lowland forest stands on Acrisol soil (at 0–180 m elevation in Jambi) measured once (11.6 µg
$N_2O$-N $m^{-2}$ $h^{-1}$; Ishizuka et al., 2005). However, soil $N_2O$ fluxes from our forests were lower
than those reported from submontane and montane forests on Cambisol soil (at 450–1160 m
elevation in Sulawesi) with six monthly measurements and comparable spatial replication (25





µg $N_2O$-N $m^{-2}$ $h^{-1}$; Veldkamp et al., 2008) and from a lowland forest on Ferralsol soil (at 100
m elevation in Jambi) with 13 monthly measurements (19.8 µg $N_2O$-N $m^{-2}$ $h^{-1}$; Aini et al.,
2015). In contrast, our values were higher than those reported for two lowland forests on
Ferralsol soil (at approximately 100 m elevation in Jambi) with nine monthly measurements
(3.0 µg $N_2O$-N $m^{-2}$ $h^{-1}$; Ishizuka et al., 2002). Since the studies from the montane forests were
conducted on less weathered soils and the studies from the same region by Ishizuka et al.
(2002, 2005) and Aini et al. (2015) have less temporal or spatial replication, their values
should be carefully related to our measured fluxes.
Soil NO fluxes from Southeast Asian lowland forests are not reported so far. Our
measured NO fluxes from the forest soils (Table 1) tended to be lower than those reported for
lowland forests in Latin America with soils ranging from less weathered Cambisols to highly
weathered Acrisols and Ferralsols (3.2 µg NO-N $m^{-2}$ $h^{-1}$, Corre et al., 2014; 10.3 NO-N $m^{-2}$ $h^{-1}$
, Davidson et al., 2004; 88.0–90.0 µg NO-N $m^{-2}$ $h^{-1}$, Keller et al., 2005; 17.4 µg NO-N $m^{-2}$ $h^{-1}$
, Verchot et al., 1999). There are only two studies that reported soil NO fluxes from montane
forests on Cambisol soils in Sulawesi, Indonesia (Purbopuspito et al., 2006, Veldkamp et al.,
2008). Our measured soil NO fluxes were comparable with the values reported for montane
forests at ≥ 1800 m elevation (1.9–2.1 µg NO-N $m^{-2}$ $h^{-1}$; Purbopuspito et al., 2006) but lower
than those reported for (pre)montane forests at lower elevations (5.5 µg NO-N $m^{-2}$ $h^{-1}$ at 1190
m, Purbopuspito et al., 2006; 12.0  µg NO-N $m^{-2}$ $h^{-1}$ at 450–1160 m, Veldkamp et al., 2008).
Although it is known that tropical forest soils are the largest natural source of $N_2O$ and
produce considerable amounts of NO, our measurements from these lowland forests in Jambi,
Indonesia on highly weathered Acrisol soils showed generally low soil N-oxide fluxes.
In contrast to our first hypothesis, soil N-oxide fluxes from the reference land uses
were comparable between loam and clay Acrisol landscapes. This is possibly due to the





generally low soil N availability in these sites, as indicated by their lower gross N
mineralization rates (Allen et al., 2015) compared, for example, to the less weathered
Cambisol and Nitisol soils in a lowland forest of Panama (Corre et al., 2010). Soil N-oxide
fluxes are largely controlled, first, by the magnitude of soil N availability, as depicted in the
HIP conceptual model (Davidson et al., 2000). This influence of soil N availability on N-
oxide fluxes was illustrated by the positive correlations of soil N-oxide fluxes with soil $NO_3^-$
contents (Table 3). Across landscapes, this first level of control was also corroborated by the
positive correlations of annual soil $N_2O$ fluxes with gross nitrification rates, and within each
landscape by the negative correlation with the soil C:N ratio (clay Acrisol landscape) and by
the positive correlation with microbial C (loam Acrisol landscape) (see Sect. 3.4). Our
findings were consistent with those from other tropical soils, illustrating that soil N-oxide
fluxes across or within sites are controlled by soil N availability as expressed in various
indexes such as soil $NO_3^-$ contents (Keller and Reiners, 1994; Müller et al., 2015),
nitrification rates (Davidson et al., 2000) and soil C:N ratio (Breuer et al., 2000).
Moreover, we attributed the low soil NO fluxes and the dominance of $N_2O$ (Table 1)
in our sites to the second level of control of N-oxide fluxes - soil aeration status (HIP model;
Davidson et al., 2000). The ratio of $N_2O$ to NO  is expected to increase when WFPS exceeds
60 % as low soil aeration favors $N_2O$ production by denitrification and nitrification processes
(Davidson et al., 2000). WFPS in the reference land uses were ≥ 60 % (Appendix Table A2,
except in jungle rubber of the loam Acrisol with 54 % WFPS). Hence, it was not surprising
that our measured soil NO fluxes were close to zero or showed net consumption (Table 1); the
high WFPS may have led to NO reduction to $N_2O$ (Conrad, 1996; Pilegaard, 2013). This was
supported by the negative correlation between soil NO fluxes and WFPS (Table 3).
Furthermore, increased concentrations of NO in the atmosphere due to biomass burning in
this region (Field et al., 2009; Levine, 1999) may have resulted in a net NO consumption (not



only in the reference land uses but also in the converted land uses; Table 1) since increased
ambient NO concentration could enhanced soil NO uptake (Conrad, 1994). In summary, soil
NO fluxes from the reference land uses were of minor importance compared to soil $N_2O$
fluxes. However, if droughts will occur more frequently or extremely in this region (Lestari et
al., 2014), soil NO fluxes might become important.
**4.2 Land-use change effects on soil $N_2O$ and NO fluxes**
Soil $N_2O$ fluxes from our rubber plantations (Table 1) were comparable to fluxes from a
rubber plantation on Ferralsol soil (at approximately 110 m elevation in Peninsular Malaysia)
with eight measurements during 1.5-year period (7.8 µg $N_2O$-N $m^{-2}$ $h^{-1}$; Yashiro et al., 2008)
and slightly higher than fluxes reported from a rubber plantation on a lateritic soil (at 580 m
elevation in Xishuangbanna, China) with only two months of sampling (4.1 µg $N_2O$-N $m^{-2}$ $h^{-1}$;
Werner et al., 2006). Studies from the same region (Jambi, Indonesia) report lower soil $N_2O$
fluxes from one rubber plantation on Ferralsol soil (at approximately 100 m elevation) with
nine monthly measurements (0.7 µg $N_2O$-N $m^{-2}$ $h^{-1}$; Ishizuka et al., 2002) as well as higher
fluxes from five rubber plantations on Acrisol soils (at 70–280 m elevation) with only one-
time measurement (20.6 µg $N_2O$-N $m^{-2}$ $h^{-1}$; Ishizuka et al., 2005) and from one rubber
plantation on Ferralsol soil (at 100 m elevation) with 13 monthly measurements (11.6 µg
$N_2O$-N $m^{-2}$ $h^{-1}$; Aini et al., 2015). Soil $N_2O$ fluxes from our oil palm sites were in the same
order of magnitude as those reported from three oil palm plantations on Acrisol soils (at 70–
110 m elevation) with only one-time sampling (15.1 µg $N_2O$-N $m^{-2}$ $h^{-1}$; Ishizuka et al., 2005)
and from one oil palm plantation on Cambisol soil (at 70 m elevation) with 13 monthly
measurements (11.9 µg $N_2O$-N $m^{-2}$ $h^{-1}$; Aini et al., 2015), whereby both studies were also
conducted in Jambi, Indonesia. However, soil $N_2O$ fluxes from our oil palm sites were higher



compared to fluxes reported from one oil palm plantation on Ferralsol soil (at approximately
110 m elevation) in Peninsular Malaysia with eight measurements during 1.5-year period
(-0.1 µg $N_2O$-N $m^{-2}$ $h^{-1}$; Yashiro et al., 2008). Soil NO fluxes have never been reported from
rubber or oil palm plantations. Our present study provides the first soil N-oxide flux
measurements from these land uses with sufficient temporal coverage and spatial replications
at the landscape scale.
In contrast to our second hypothesis, soil N-oxide fluxes were comparable among land
uses (except for soil NO fluxes between rubber and jungle rubber in the loam Acrisol
landscape as discussed below), even with the observed decreases in soil mineral N levels
among land uses (i.e., generally lower $NH_4^+$ and $NO_3^-$ levels in rubber plantations than in the
reference land uses at both landscapes; Appendix Table A2). In the same study sites, Allen et
al. (2015) found differences in other indices of soil N availability with land-use change,
particularly in the clay Acrisol landscape: microbial C and N, gross N mineralization and
$NH_4^+$ immobilization rates decrease with conversion of forest to rubber or oil palm
plantations. N-oxide emissions generally account only a small fraction of soil available N
(e.g., $N_2O$ + NO emissions comprise 0.03 % of gross N mineralization rates in a lowland
forest on Cambisol and Nitisols soils in Panama; Corre et al., 2014). In our present study, the
reference land uses on highly weathered Acrisol soils have low soil N availability and their
conversion to these plantations further decreases the soil N-cycling rates (Allen et al., 2015).
Hence, we reason that we did not detect differences in N-oxide fluxes with land-use
conversion to rubber and oil palm plantations because we started with low soil N availability
and low N-oxide emissions and any changes were probably too small to detect statistically.
The temporal pattern of soil $N_2O$ fluxes in the converted land uses were also controlled by
soil $NO_3^-$ contents (Table 3), emphasizing the first level of control of soil N availability on
soil $N_2O$ fluxes (HIP model; Davidson et al., 2000). Across landscapes, the correlations of



annual soil $N_2O$ fluxes from these converted land uses with sand contents (see Sect. 3.4) also
suggested the indirect influence of soil texture on water holding capacity, or conversely soil
aeration status, which is the second level of control on soil $N_2O$ fluxes (HIP model).
Consequently, in terms of N-oxide emissions, this footprint of smallholder oil palm and
rubber plantations was similar to the original land uses. However, this picture might change
with increasing usage of N fertilizer (see Sect. 4.3).
The lower soil NO fluxes in rubber compared to jungle rubber in the loam Acrisol
(Table 1) partly supports our second hypothesis. These differences might be related to the low
WFPS and the higher soil $NO_3^-$ contents in jungle rubber (Appendix Table A2), which could
favor the relatively high soil NO emissions; this was also supported by the opposing
correlations of soil NO with $NO_3^-$ and WFPS (Table 3). Additionally, the low soil NO fluxes
from rubber plantations could be the result of the effect of monoterpenes, produced by rubber
trees, which reduce nitrification in soil (Wang et al., 2007; White, 1991). This is supported by
low gross nitrification rates (measured in the same plots by Allen et al., 2015), low soil $NO_3^-$
contents (Appendix Table A2) and consequently low soil NO fluxes in rubber plantations
(Table 1).
**4.3 Soil management effects on soil $N_2O$ and NO fluxes from oil palm plantations**
N fertilizer application, a commonly employed soil management in oil palm plantations (e.g.,
Allen et al., 2015; Hassler et al., 2015), increases N-oxide emission for a relatively short
period (e.g., Koehler et al. 2009). Our findings show that these fertilizer-induced N-oxide
emissions were mainly limited to the small area around the palm base where fertilizer is
commonly applied (4 % of the area in a hectare) and that N-oxide emissions peaked within 3
weeks (Figs. 2 and 3). These N-fertilizer induced $N_2O$ fluxes of 6–21 % of the annual soil
$N_2O$ fluxes were similar in magnitude as the standard errors of the annual fluxes (estimated





from the monthly measurements; Table 1). Thus, inclusion of these N-induced emissions in
our annual estimates did not result in statistically significant effects of land-use change.
The percentages of soil $N_2O$ and NO fluxes to the applied N fertilizer rate were
smaller than those reported from other agricultural land uses in humid tropical regions (6.4–
8.6 %; Veldkamp and Keller, 1997; Veldkamp et al., 1998). Usually the percentage of soil N-
oxide emissions to applied N fertilizer rate increases with increasing N fertilization rates
(Hoben et al., 2011; Pennock and Corre, 2001). Since the fertilization rates in our studied
smallholder oil palm plantations were lower compared to the fertilization rates in these other
studies (with N fertilization rates ranging from 300–360 kg N ha$^{-1}$ yr$^{-1}$), our quantified N-
oxide loss from N fertilizer were also low. The higher soil $N_2O$ fluxes in the large-scale oil
palm plantation PTPN VI, although not statistically different from the smallholder plantations
(Table 1), could be attributed to its high N fertilization rate (196 kg N ha$^{-1}$ yr$^{-1}$). Summing the
N-induced N-oxide fluxes and the annual soil N-oxide emissions based on the monthly
measurements (Table 1), these values from the smallholder plantations were still lower than
the annual flux from the large-scale plantation (Table 1). Based on our finding that soil $N_2O$
fluxes following fertilizer application (chamber location b) were higher in the clay than loam
Acrisol landscapes (most likely due to higher WFPS in the clay (61 ± 8 %) than loam Acrisol
(27 ± 3 %) during this measurement period), soil N-oxide fluxes from large-scale plantations
on clay soils could be even higher than what we measured here from a large-scale plantation
on a loam soil. Our findings reinforced the need to quantify these climate-relevant N-oxide
gases in large-scale plantations, which constitute ~50 % of the land area under oil palm
plantation in whole of Sumatra (BPS, 2014).
Temporal patterns in soil N-oxide fluxes following fertilizer application were also
controlled by soil N availability, as reflected by their positive correlations with soil $NH_4^+$
and/or $NO_3^-$ contents (Table 4). The pulse application of N fertilizer provide temporary



surplus of mineral N that was lost via gaseous emission and leaching (Kurniawan, 2016), and
with time following fertilizer application such effect diminished as the mineral N is
incorporated into the soil N-cycling processes (Allen et al., 2015). The positive correlation
between soil $N_2O$ fluxes and WFPS (i.e., chamber locations a and b in site 1 of the loam
Acrisol; Table 4) and the negative correlation between soil NO fluxes and WFPS (i.e.,
chamber location b in site 3 of the clay Acrisol landscape; Table 4) again attested that when
the first level of control (soil N availability) was favorable (i.e., high soil mineral N contents
in these fertilized chamber locations) the control of soil moisture on aeration status was
enhanced, as such correlation was not seen in the unfertilized area (chamber location c) or in
the monthly measured fluxes (Tables 3 and 4). These correlations indicated that following
fertilizer application soil NO fluxes decreased whereas soil $N_2O$ fluxes increased with
increases in WFPS. In site 3 of the loam Acrisol, the seemingly contradicting negative
correlation of soil $N_2O$ fluxes with WFPS (Table 4) was only because there was a decreasing
WFPS following fertilizer application with concurrently increasing soil mineral N contents -
the latter dominantly driving the increases in soil $N_2O$ fluxes (i.e., positive correlations with
$NH_4^+$ and $NO_3^-$; Table 4). In summary, the short-term effect of fertilization also depicted the
two levels of controls on soil N-oxide fluxes as exemplified in the HIP model.
**5 Conclusions**
Our study provides the first spatially replicated study with a full year of measurements of soil
$N_2O$ fluxes and the first reported soil NO fluxes from this region of hotspot of land-use
conversion for globally important tree cash crops. In contrast to our first hypothesis, soil
texture, through its role on soil fertility, did not directly affect soil N-oxide fluxes (as shown
by the comparable fluxes between landscapes with soil textural differences) but influenced the
landscape-scale pattern of annual soil $N_2O$ fluxes in the converted land uses (i.e., negative



correlation between annual $N_2O$ fluxes and sand content) most likely through its role on soil
moisture availability. The generally low soil N-oxide fluxes from the reference land uses were
due to the low soil N availability in these highly weathered Acrisol soils (Allen et al., 2015).
Forest or jungle rubber conversion to rubber and oil palm by smallholders also did not show
significant changes in soil N-oxide fluxes, except for the decrease in soil NO fluxes in rubber
plantations and for the short-term pulse of soil N-oxide fluxes following fertilizer application
in oil palm plantations. These partly support our second hypothesis. Using a conservative
estimate of N-oxide ($N_2O$ + NO) loss from the applied N fertilizer (average of 0.5 % from the
loam and clay Acrisol landscapes), and a conservative average N fertilization rate across
smallholder and large-scale plantations of 100 kg N $ha^{-1}$ $yr^{-1}$, with the total land area of oil
palm in Jambi province of 721000 ha (BPS, 2014), we estimated an annual soil N-oxide
emission from N fertilization of 360500 kg N $yr^{-1}$. The N fertilization rates in our smallholder
oil palm plantations were only about one-fourth to one-half of what is commonly practiced in
large-scale industrial plantations (e.g., 130–260 kg N $ha^{-1}$ $yr^{-1}$ in Jambi, Indonesia; Pahan,
2010), and our measurements from a large-scale oil palm plantation PTPN VI showed high
soil N-oxide fluxes. To improve estimate of soil N-oxide fluxes at regional level, future
studies should focus on large-scale plantations (which constitute 38 % of oil palm land area in
Jambi province; BPS, 2014) with frequent measurements during 2 months following fertilizer
application, and particularly during wet season for $N_2O$ flux measurements and during dry
season for NO flux measurements.
**Data availability**
The underlying research data of this study is deposited at the EFForTS-IS data repository
(https://efforts-is.uni-goettingen.de), an internal data exchange-platform, which is accessible
for SFB 990 members only. Based on data sharing agreement within the SFB 990, these data



are currently not publicly accessible but will be made available through a written request to
the senior authors.
**Competing interests**
The authors declare that they have no conflict of interest.
*Acknowledgments.* We thank the village leaders, local plot owners, PT REKI, PTPN VI, and
Bukit Duabelas National Park for granting us access and use of their properties. This study
was financed by the Deutsche Forschungsgemeinschaft (DFG) as part of the project A05
(SFB 990/2) in the framework of the German-Indonesian Collaborative Research Center 990:
Ecological and Socioeconomic Function of Tropical Lowland Rainforest Transformation
Systems. We are especially grateful to our Indonesian assistants, Edward Januarlin Siahaan,
Nelson Apriadi Silalahi, Ardi, Fahrurrozy, Edi, Bayu Puja Kesuma, Basri, Darwis and Suriana
as well as all the rangers of the protected forest areas. We also acknowledge project A03 for
helping part of the gas sampling in PTPN VI, both A03 and the Indonesian Meteorological,
Climatological and Geophysical Agency for climatic data, as well as the other members of
project A05 (Allen et al., 2015) for the soil physical and biochemical data (Appendix Table
A1), and also B04 (Kotowska et al., 2015) and B06 (Rembold et al., unpublished data) for
providing vegetation data. We thank Norman Loftfield, Oliver van Straaten, Andrea Bauer,
Kerstin Langs and Martina Knaust (Georg-August University Göttingen, Germany) for their
assistance with laboratory analyses. This study was conducted using the research permits
(210/SIP/FRP/SM/VI/2012 and 45/EXT/SIP/FRP/SM/V/2013) from the Ministry of Research
and Technology of Indonesia (RISTEK), and the collection permits
(2703/IPH.1/KS.02/XI/2012 and S.13/KKH-2/2013) from the Indonesian Institute of Sciences
(LIPI) and the Ministry of Forestry (PHKA).



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



**Table 1.** Mean (±SE, $n$ = 4 sites) soil $N_2O$ and NO fluxes and annual soil $N_2O$ fluxes from different land uses within each landscape in Jambi, Sumatra, Indonesia, measured monthly from December 2012 to December 2013. Means followed by different lowercase letters indicate significant differences among land uses within each landscape and different capital letters indicate significant differences between landscapes within each land use (linear mixed-effect models with Fisher's LSD test at $P \le 0.09$). For soil NO fluxes in the clay Acrisol, forest was excluded in the comparison among land uses because its monthly measurements was only carried out in two sites due to road inaccessibility with the NO-measuring instrument in the other two sites. Annual soil $N_2O$ fluxes were not statistically tested for differences between landscapes or land uses since these annual values are trapezoidal extrapolations. For smallholder oil palm plantations, values in italics are the fertilizer-induced annual soil $N_2O$ fluxes (see Sect. 2.2). In the loam Acrisol landscape, soil $N_2O$ fluxes were additionally measured in a large-scale oil palm plantation (mean±SE, $n$ = 4 replicates) from July 2014 to July 2015; these fluxes did not differ from those of smallholder plantations in the same landscape (linear mixed-effect models with Fisher's LSD test at $P$ = 0.15) due to large spatial variation (indicated by large SE).

| Land-use type | $N_2O$ fluxes ($\mu g\ N\ m^{-2}\ h^{-1}$) | NO fluxes ($\mu g\ N\ m^{-2}\ h^{-1}$) | Annual $N_2O$ fluxes ($kg\ N\ ha^{-1}\ year^{-1}$) |
|---|---|---|---|
| clay Acrisol landscape | | | |
| Forest | $12.76 \pm 5.57^{a,A}$ | $(1.70 \pm 0.32)$ | $1.03 \pm 0.41$ |
| Jungle rubber | $6.73 \pm 1.50^{a,A}$ | $-0.56 \pm 0.69^{a,A}$ | $0.62 \pm 0.14$ |
| Rubber | $5.56 \pm 2.47^{a,A}$ | $-1.00 \pm 0.15^{a,A}$ | $0.46 \pm 0.21$ |
| Oil palm (smallholder plantation) | $11.47 \pm 2.88^{a,A}$ | $-0.20 \pm 1.23^{a,A}$ | $1.01 \pm 0.25$ *$0.21 \pm 0.04$* |
| loam Acrisol landscape | | | |



| | | | |
|---|---|---|---|
| Forest | $9.77 \pm 1.46^{a,A}$ | $1.87 \pm 1.27^{ab}$ | $0.88 \pm 0.15$ |
| Jungle rubber | $14.01 \pm 6.69^{a,A}$ | $5.68 \pm 5.77^{a,A}$ | $1.19 \pm 0.57$ |
| Rubber | $8.61 \pm 2.04^{a,A}$ | $-1.16 \pm 0.49^{b,A}$ | $0.69 \pm 0.17$ |
| Oil palm (smallholder plantation) | $12.16 \pm 6.08^{a,A}$ | $0.73 \pm 0.67^{ab,A}$ | $1.13 \pm 0.53$<br>*$0.07 \pm 0.02$* |
| Oil palm (large-scale plantation) | $42.34 \pm 24.22^{a,A}$ | - | $3.26 \pm 1.73$ |



**Table 2.** Mean (±SE, $n$ = 3 oil palm trees) soil $N_2O$ and NO fluxes from three chamber
locations during a fertilization in three (for $N_2O$) or one (for NO) smallholder oil palm
plantation within each landscape, measured 6 to 11 times during 3–8.5 weeks following
fertilization. Means followed by different letters indicate significant differences among
chamber locations within each site (linear mixed-effect models with Fisher's LSD test at
$P \leq 0.05$). Chamber locations a, b and c were placed at 0.3 m, 0.8 m, and 4–4.5 m,
respectively, from each of the three trees in each oil palm plantation site. Smallholders
fertilized around the base of each tree at about 0.8–1 m from the tree base, and thus chamber
location b was on this fertilized area and chamber location c served as the reference chamber
not receiving any fertilizer. The same fertilization rate and form were used as the smallholders
applied in these plantations (see Sect. 2.2).

| Oil palm site | Chamber location | $N_2O$ fluxes (µg N m$^{-2}$ h$^{-1}$) | NO fluxes (µg N m$^{-2}$ h$^{-1}$) |
|---|---|---|---|
| | | clay Acrisol landscape | |
| 1 | a | 156.66 ± 86.76[b] | - |
| | b | 910.11 ± 410.00[a] | - |
| | c | 6.93 ± 3.30[c] | - |
| 2 | a | 130.62 ± 34.62[b] | - |
| | b | 692.74 ± 144.10[a] | - |
| | c | 9.87 ± 3.01[c] | - |
| 3 | a | 45.49 ± 3.73[b] | 4.74 ± 1.74[b] |
| | b | 1280.95 ± 486.67[a] | 535.29 ± 194.46[a] |
| | c | 1.14 ± 1.64[c] | 1.50 ± 1.46[b] |





| Oil palm site | Chamber location | N$_2$O fluxes (µg N m$^{-2}$ h$^{-1}$) | NO fluxes (µg N m$^{-2}$ h$^{-1}$) |
|---|---|---|---|
| | | loam Acrisol landscape | |
| 1 | a | 33.46 ± 9.76[b] | - |
| | b | 133.36 ± 34.90[a] | - |
| | c | 11.82 ± 6.08[b] | - |
| | | | |
| 2 | a | 129.74 ± 46.19[a] | 46.17 ± 19.63[b] |
| | b | 205.31 ± 24.17[a] | 157.12 ± 35.67[a] |
| | c | 7.89 ± 4.78[b] | 0.66 ± 0.30[b] |
| | | | |
| 3 | a | 5.17 ± 1.04[b] | - |
| | b | 104.53 ± 81.90[a] | - |
| | c | 3.68 ± 1.74[b] | - |





**Table 3.** Pearson correlation coefficients between soil $N_2O$ flux ($n = 48$; µg N $m^{-2}$ $h^{-1}$), soil
NO flux ($n = 16$; µg N $m^{-2}$ $h^{-1}$), water-filled pore space (WFPS; %, top 0.05 m depth), soil
temperature (°C, top 0.05 m depth) and extractable mineral N (mg N $kg^{-1}$, top 0.05 m depth)
across landscapes for the reference and converted land uses. Correlation was conducted using
the means of the four replicate plots per land use on each monthly measurement from
December 2012 to December 2013 (soil $N_2O$ fluxes) and March 2013 to September 2013 (soil
NO fluxes).

| Land-use type | Variable | WFPS | Soil temp. | $NH_4^+$ | $NO_3^-$ |
|---|---|---|---|---|---|
| Reference land uses (forest and jungle rubber) | Soil $N_2O$ flux | -0.21 | -0.09 | -0.23 | 0.38[c] |
| | Soil NO flux | -0.74[c] | -0.15 | -0.48[a] | 0.69[c] |
| Converted land uses (rubber and oil palm) | Soil $N_2O$ flux | 0.11 | 0.30[b] | 0.23 | 0.37[c] |
| | Soil NO flux | -0.05 | 0.09 | -0.05 | 0.23 |

[a]$P \leq 0.09$, [b]$P \leq 0.05$, [c]$P \leq 0.01$.





**Table 4.** Pearson correlation coefficients ($n$ = 6–11 measurements following fertilization)
between N-oxide fluxes ($\mu$g N m$^{-2}$ h$^{-1}$), water-filled pore space (WFPS; %, top 0.05m depth)
and extractable mineral N (mg N kg$^{-1}$, top 0.05 m depth), measured at different chamber
locations (a, b and c were at 0.3 m, 0.8 m (fertilized area) and 4–4.5 m, respectively, from
each of the three trees in each smallholder oil palm plantation). Correlation was conducted
using the means of the three replicate trees per chamber location.

| Oil palm plantation site | Chamber location | Variable | WFPS | $NH_4^+$ | $NO_3^-$ |
|---|---|---|---|---|---|
| clay Acrisol landscape | | | | | |
| 1 | a | | 0.55 | 0.88[b] | 0.46 |
| ($n$ = 6 measurements) | b | Soil $N_2O$ flux | 0.57 | -0.22 | -0.31 |
| | c | | 0.37 | -0.64 | -0.44 |
| 2 | a | | 0.11 | 0.93[c] | 0.95[c] |
| ($n$ = 11 measurements) | b | Soil $N_2O$ flux | 0.08 | 0.05 | -0.06 |
| | c | | 0.09 | -0.44 | -0.45 |
| 3 | a | | -0.19 | 0.10 | 0.09 |
| ($n$ = 10 measurements) | b | Soil $N_2O$ flux | 0.05 | 0.86[c] | 0.85[c] |
| | c | | -0.32 | 0.06 | -0.44 |
| 3 | a | | -0.34 | 0.44 | 0.48 |
| ($n$ = 10 measurements) | b | Soil NO flux | -0.61[a] | 0.10 | -0.04 |
| | c | | 0.59[a] | -0.14 | -0.13 |
| loam Acrisol landscape | | | | | |
| 1 | a | | 0.96[c] | -0.18 | 0.03 |
| ($n$ = 6 measurements) | b | Soil $N_2O$ flux | 0.78[a] | 0.61 | -0.40 |
| | c | | -0.06 | -0.29 | <0.01 |
| 2 | a | | -0.55 | 0.71[b] | -0.03 |
| ($n$ = 9 measurements) | b | Soil $N_2O$ flux | 0.35 | -0.20 | 0.89[c] |
| | c | | 0.34 | <0.01 | -0.35 |





| | | | | | |
|---|---|---|---|---|---|
| 3 | a | | -0.68[b] | 0.67[b] | 0.62[b] |
| ($n = 11$ measurements) | b | Soil $N_2O$ flux | -0.27 | -0.2 | 0.57[a] |
| | c | | 0.36 | 0.19 | 0.06 |
| | | | | | |
| 2 | a | | -0.07 | 0.18 | -0.27 |
| ($n = 9$ measurements) | b | Soil NO flux | 0.07 | -0.11 | 0.96[c] |
| | c | | -0.16 | 0.12 | -0.23 |

[a]$P \leq 0.09$, [b]$P \leq 0.05$, [c]$P \leq 0.01$.





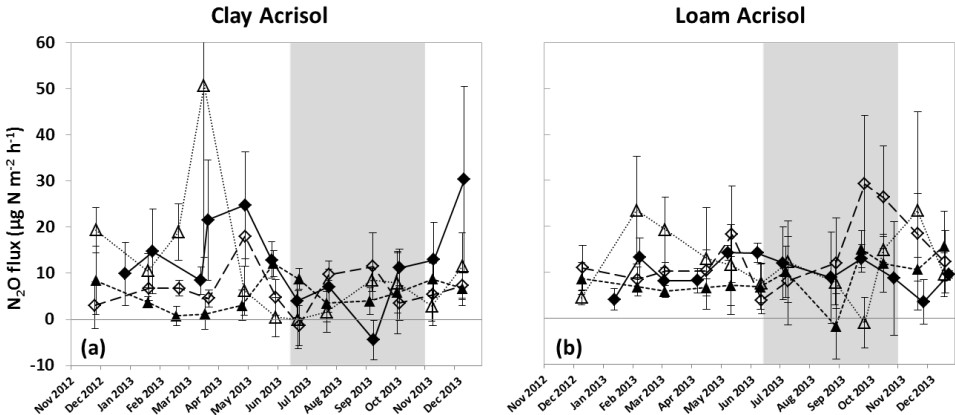

**Figure 1.** Mean (±SE, *n* = 4 sites) soil $N_2O$ fluxes from forest (♦), jungle rubber (◊),
rubber (▲) and oil palm (△), located within the clay (**a**) and loam Acrisol (**b**) landscapes in
Jambi, Sumatra, Indonesia. Measurements were carried out monthly from December 2012 to
December 2013; grey shadings mark the dry season.



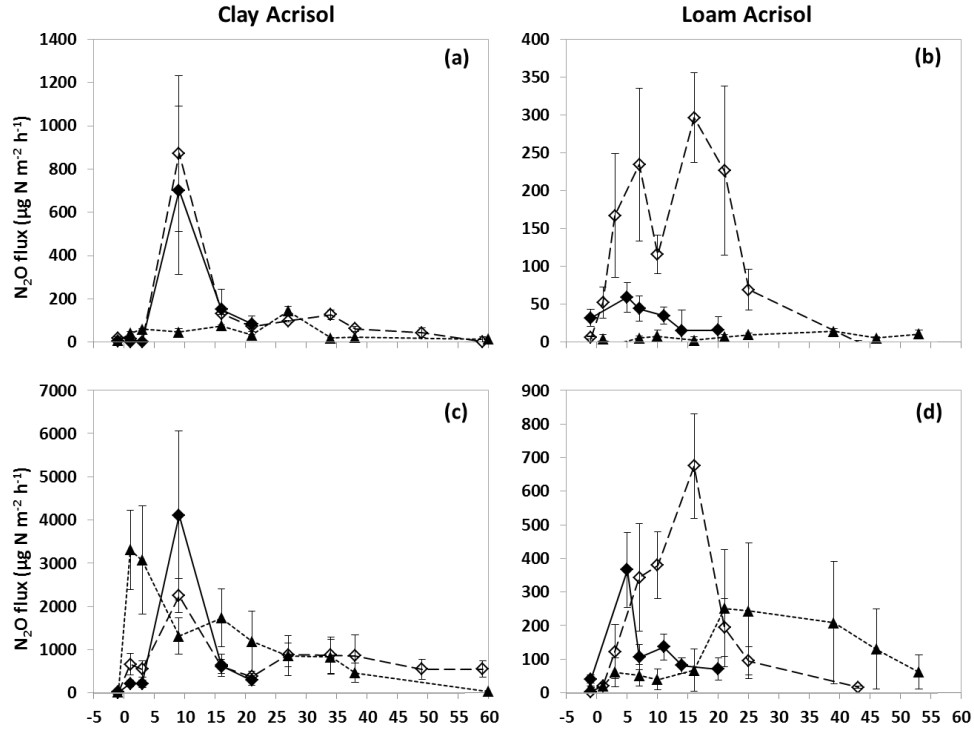

**Figure 2.** Mean ($\pm$SE, $n$ = 3 oil palm trees) soil $N_2O$ fluxes during a fertilization in

smallholder oil palm plantations 1 (♦), 2 (◇) and 3 (▲) in the clay (**a** and **c**) and loam Acrisol

(**b** and **d**) landscapes. Smallholders fertilized around the base of each tree at about 0.8–1 m

from the tree base. Fluxes were measured at 0.3 m from the tree base (**a** and **b**) and at 0.8 m

on the fertilized location (**c** and **d**) with the same rate and form that smallholders used (see

Sect. 2.2).



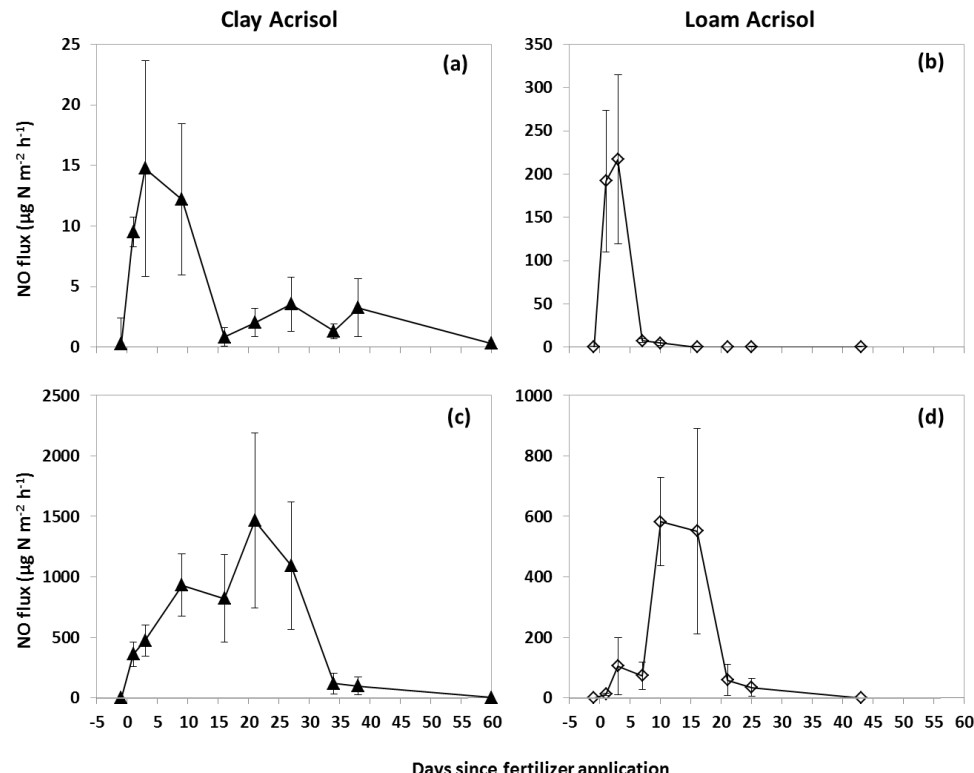

**Figure 3.** Mean (±SE, $n$ = 3 oil palm trees) soil NO fluxes during a fertilization in a

smallholder oil palm plantation in the clay (**a** and **c**) and loam Acrisol (**b** and **d**) landscapes.

Smallholders fertilized around the base of each tree at about 0.8–1 m from the tree base.

Fluxes were measured at 0.3 m from the tree base (**a** and **b**) and at 0.8 m on the fertilized

location (**c** and **d**) with the same rate and form that smallholders used (see Sect. 2.2).





**Table A1.** Mean (±SE, $n = 4$ sites) soil physical and biochemical characteristics in the top 0.10 m depth (except sand content with $n = 3$ sites) from different land uses within each landscape in Jambi, Sumatra, Indonesia. Means followed by different lowercase letters indicate significant differences among land uses within each landscape and different capital letter indicate significant differences between landscapes within each land use (linear mixed-effect models with Fisher's LSD test at $P \leq 0.05$ and marginally significant at $*P \leq 0.09$). These soil characteristics were reported by Allen et al. (2015), except for the sand content (Allen et al., unpublished data).

| Soil characteristics | Land-use type | | | |
| --- | --- | --- | --- | --- |
| | Forest | Jungle rubber | Rubber | Oil palm |
| **Clay Acrisol landscape** | | | | |
| Sand (%) | $36 \pm 11^{a}$ | $27 \pm 20^{a}$ | $35 \pm 7^{a}$ | $11 \pm 2^{a,B*}$ |
| Soil C:N ratio | $13.1 \pm 1.3^{a}$ | $13.0 \pm 0.3^{a}$ | $14.3 \pm 0.6^{a,A}$ | $13.5 \pm 0.2^{a}$ |
| Microbial C (mg C kg$^{-1}$) | $1048 \pm 201^{a*,A}$ | $922 \pm 223^{ab*}$ | $561 \pm 61^{c*}$ | $617 \pm 112^{bc*}$ |
| Gross nitrification (mg N kg$^{-1}$ day$^{-1}$) | $0.9 \pm 0.3^{a}$ | $1.0 \pm 0.2^{a}$ | $0.7 \pm 0.2^{a}$ | $2.0 \pm 0.8^{a}$ |
| **Loam Acrisol landscape** | | | | |
| Sand (%) | $39 \pm 8^{a}$ | $42 \pm 19^{a}$ | $26 \pm 13^{a}$ | $43 \pm 14^{a,A*}$ |
| Soil C:N ratio | $14.3 \pm 0.2^{a}$ | $13.7 \pm 0.8^{a}$ | $11.7 \pm 0.7^{b,B}$ | $12.5 \pm 0.5^{ab}$ |
| Microbial C (mg C kg$^{-1}$) | $514 \pm 48^{a,B}$ | $578 \pm 45^{a}$ | $461 \pm 58^{a}$ | $403 \pm 24^{a}$ |
| Gross nitrification (mg N kg$^{-1}$ day$^{-1}$) | $1.9 \pm 0.4^{a}$ | $0.9 \pm 0.2^{a}$ | $0.9 \pm 0.2^{a}$ | $1.2 \pm 0.5^{a}$ |



**Table A2.** Mean (±SE, $n$ = 4 sites) soil water-filled pore space (WFPS) and extractable
mineral N in the top 0.05 m depth for different land uses within each landscape in Jambi,
Sumatra, Indonesia, measured monthly from December 2012 to December 2013. Means
followed by different lowercase letters indicate significant differences among land uses within
each landscape and different capital letters indicate significant differences between landscapes
within each land use (linear mixed-effect models with Fisher's least significant difference
(LSD) test at $P \le 0.05$). These soil characteristics were reported by Hassler et al. (2015).

| Land-use type | WFPS (%) | $NH_4^+$ (mg N kg$^{-1}$) | $NO_3^-$ (mg N kg$^{-1}$) |
|---|---|---|---|
| clay Acrisol landscape | | | |
| Forest | 72.97 ± 12.31[a,A] | 6.99 ± 1.03[a,A] | 2.15 ± 0.36[a,A] |
| Jungle rubber | 86.74 ± 5.93[a,A] | 7.33 ± 0.21[a,A] | 0.23 ± 0.06[b,B] |
| Rubber | 61.49 ± 7.41[a,A] | 4.25 ± 0.23[b,A] | 0.05 ± 0.01[b,B] |
| Oil Palm | 74.03 ± 7.28[a,A] | 5.80 ± 0.64[a,A] | 0.81 ± 0.49[b,] |
| loam Acrisol landscape | | | |
| Forest | 63.97 ± 3.30[a,A] | 5.94 ± 0.40[a,A] | 0.61 ± 0.15[ab,B] |
| Jungle rubber | 53.86 ± 3.70[a,B] | 5.64 ± 0.28[a,B] | 1.25 ± 0.63[a,A] |
| Rubber | 72.58 ± 5.73[a,A] | 4.14 ± 0.57[b,A] | 0.12 ± 0.02[b,A] |
| Oil Palm | 59.04 ± 6.74[a,A] | 4.20 ± 1.10[b,B] | 0.60 ± 0.36[ab,B] |

