# Peer review of "Soil nitrogen oxide fluxes from lowland forests converted to"

_Biogeosciences, 2016_

## Referee Comment (RC1) · Anonymous Referee #1 · 19 Nov 2016

The authors present measurement results for nitrous oxide and nitric oxide emissions from natural forest and important land-uses (rubber, oil plant) for a region in Sumatra, Indonesia. As these measurements are labour and/ or capital intensive they are still scarce especially for (remote) regions of the tropics and sub-tropics. In light of the strong temporal and spatial variability of the soil-atmosphere exchange of these gases and their importance for the greenhouse gas balance and tropospheric chemistry, new field measurements are of great merit to the scientific community and publication should be promoted whenever possible. Considering the logistics and associated costs the sampling intervals are sufficient but a more frequent sampling routine and/ or (semi-)automated sampling procedures would have helped to better cover the, often dynamic, gas exchange (but a monthly sampling interval is generally reasonable to illustrate seasonal dynamics). However, this rather low sampling frequency leads to poor (temporal) replication (as illustrated by the substantial standard errors in many sampling times) even if the spatial sampling design is sound.

A general problem I have with this manuscript is the length of many sections and the wordiness of many paragraphs. The abstract alone comes in at ∼430 words and could be substantially shortened (no need to describe site replication for instance). The "Materials and Methods" section for instance is extremely long (9.5 pages) and should be streamlined.

In contrast, the discussion in particular would benefit from greater detail (and discussion with results from other regions of the world). Also to me, a clear site/ replicate nomenclature would better guide the reader through the text as the full measurement setup is rather complex (two soil landscapes, 4 land uses, 3 chamber positions at each site, 4 replicates). For instance, if the authors could define some site abbreviations (e.g., reference land uses: F (forest), JR (jungle rubber); converted uses: RP (rubber plantation), PP (oil palm plantation), they could simple use to those instead to repeat the site attributes or be overly descriptive. This is also true for the naming of the three within-site chamber positions (currently: a, b, c). While their properties are described in the text, and also in the caption of table 4 it makes the digestion of the data presented unnecessary hard for the reader (maybe: F1 (fertilised area position 1 / 0.3m from stem), F2 (fertilised area position 1 (0.8m from stem), NF (non fertilised: 4.5m from stem). In lengthy paragraphs it is easy to get lost and scramble to read up what i.e. position b represents (same for the reference to the proposed hypothesis').

Furthermore, I feel that reorganising and cleaning the tables would help the better digest the main results presented. Table A1 & A2 should be combined and added to the main text.

The figures are appropriate, but could also be improved, too (see detailed comments).

Please also note the supplement to this comment:
http://www.biogeosciences-discuss.net/bg-2016-357/bg-2016-357-RC1-
supplement.pdf

—————————————————————

[Figure]

**Supplement:**

**Detailed response**

In the following I'd like to suggest some chances to the tables (often admittedly personal preference):

**Tables.**

Table 1.
- Shorten the caption (16 lines of description).
- Also, a column with of number of samples (n) would help the reader to assess the robustness of the given average emissions.
- I would suggest to round to the first decimal to reduce visual clutter (esp. with the group identifiers present in the table)

Table 2.
- The chamber location identifiers a, b, c do not help the reader. Either also identify the distance to tree in the table or use descriptive abbreviations?
- Again, indicate the number of measurements considered
- Given the lack of NO data for 4 of the 6 sampled sites, maybe another organization would be better?

    For instance:

    Table 2a (N2O) – columns:

    Oil palm site / CH pos / N2O (clay Acrisol) / N2O (loam Acrisol)

    Table 2b (NO) – columns (with oil palm site index given as NO column identifier (?)):
    CH pos / NO (Acrisol) / NO (loam Acrisol)

Table 3./4.
- They could go into the appendix
- Table 4 should be split into N2O and NO data (see Table 2)

Table A1 & A2.
- Shorten caption
- Combine A1 & A2  into one table and add it into main document as a site description/ referenece for the the reader
- Round WFPS, NH4 and NO3 to one digit to reduce clutter
- This might be personal preference, but maybe remove the significance letters, too (they make the table really hard to read, also almost all entries in A1 have a lowercase 'a', maybe only label when they differ?; and important differences can be discussed in the manuscript).

**Figures.**
Some scale modification and additional labels would make the figures easier to read.

Fig 2.
- Matching scales would help the reader (at least for groups; a) & c) and b) & d))
- Add Tree-base distance in the plots to guide the reader
- Add fertilizer amounts to plot or caption (instead of referring section 2.2)

Fig 3.
- See comments for figure 2 (y-axis breaks for a)&b) required)

**Detailed comments:**

| | |
|---|---|
| p5, l19: | Introduce site abbreviations that you can refer to in the text |
| p5, l22: | introduce H1 and H2 for your hypothesis so you can refer to them in your discussion |
| p6 | I would give a site property table here (basically combine A1&A2) and add soil properties – I feel a reference to Allen 2015 & Hassler 2015 for such fundamental information for the manuscript is not sufficient |
| p6, l16 | is the precip data given as SD? |
| p6, l20 | that's actually substantially higher |
| p7-8 | site & design description could be shortened substantially |
| p8 | Please work on the language in this section: I counted 'was done' 5 times in this paragraph |
| p9, l20 | give a reference for N fertilizer induced pulse emissions |
| p11 | trapezoidal rule should be explained briefly here (esp. since it's not explained in the given reference Hassler et al., 2015 either; in there is another reference to Koehler et al./ Veldkamp 2013). |
| p12, l4-l10 | This is hard to read; just give the equation |
| p13, l10 | "when necessary" – explain |
| p13, l13 | briefly remember the reader about your hypothesis H1 & H2 here |
| p13, l22 – p14, l 15 | tis is very detailed… maybe move this into the appendix/ a supplement? |
| p15, l11 | mention the reference land uses again |
| p15, l11 | "…from soils. In the clay …" |
| p15, l15 | Was this systematic? I.e., was there always one measurement (position) an outlier? |
| p16, l3-4 | give the fertilizer rates here, too |
| p16, l6 | "In the chamber position closest to the tree, soil N2O emissions…" |
| p16, l9 | There's also a peak for site 1 (but smaller) |
| p16, l18 | Due to which assumptions? Trees per ha? Avg. basal area of those trees? |
| p18, l3 | NH4 (only weak?) |
| p18, l5 | What is the temperature amplitude between the measurements? Relatively minor I suppose due to the tropical climate? |
| p18,l5-l9 | Remove this single sampling period outright since it clearly seems fertilizer-induced |
| p18, l13 -l14 | How is this possible? |
| p19, l17 | Give the range of your fluxes here for comparison |

| | |
|---|---|
| p19, l19-l25 | This is very wordy, could be shortened substantially |
| p20, l8 | What about the other literature? You only compare to reports from your specific region |
| p21, l24 | I do not get the reasoning here. Were there fires going on in the region during the measurements? |
| p22, l8 | Give the observed flux range here for better comparison, also the N application rates would help to judge the observations |
| p22, l9 | Why do you give the elevation here? This is not really a factor (110m, 580, …) |
| p22, l12 | However, the sampling there was very detailed and covered the transition period |
| p22, l15 | "nine monthly" is a bit deceptive, it's 9 single measurements, right? |
| p22 | Maybe a literature review table with relevant citations for the investigated landuses combined with your results would be appropriate here? This would also help the better interpret your results in context. |
| p23, l4-l6 | Also true, this seems unnecessary to mention here. Maybe give a half-sentence in the abstract highlighting the novelty of your NO measurements. |
| p23, l7 | remind the reader about the hypothesis again |
| p24, l21 | Isn't it expected that fertilizer-induced emissions occur at the site where fertilizer is applied?!? |
| p25, l8 | mention your fertilizer rates again for comparison |
| p25, l9 | these seem high; please give the references |
| p25, l25 | pulse application? Maybe: "the event-based application of high N rates" or something similar? |
| p26, l10 – l12 | This is most likely not true for low – medium moisture levels |
| p26, l12 – l16 | This sentence actually highlights a key problem with such an extensive sampling routine and should be discussed further |
| p26, l20 | true, although the "full year" is based on few measurements |
| p26, l22 | Name the hypothesis, the reader might have forgotten which hypothesis was which |
| p27, l7 | ditto |
| p27, l12 | change unit 'kg' to 't' |

---

## Referee Comment (RC2) · Y. A. Teh (Referee) · 23 Nov 2016

Y. A. Teh (Referee)

yateh@abdn.ac.uk

**GENERAL COMMENTS**

Data on N-oxide fluxes from oil palm ecosystems in SE Asia or elsewhere in the tropics is scarce, making it difficult to estimate or predict how existing land-use practices influence the flux of N-oxides to the atmosphere, and regionally/globally important processes such as tropospheric ozone formation, N-deposition, or climate forcing. Improving our understanding of the role played by management, soil type, environmental factors, and other control variables in modulating N-oxide fluxes from oil palm is critical if we hope to evaluate the impact of land-use change on the regional and global N cycle. This is especially important for a major land-use like oil palm, which accounts for

>13% of the tropical land area, and is expanding rapidly across the African and Latin American tropics.

The work presented here by Hassler et al. (2016) is therefore novel and timely because it helps us to start addressing these knowledge gaps. The focus of this research in one of the most heavily oil palm-dominated areas in SE Asia is also important, given that the process-level insights derived from this work can help us understand the functioning and behaviour of similar systems elsewhere in this area. The comparison among multiple land-uses, soil types, and different fertilization regimes (e.g. lower intensity small holder vs higher intensity large holder) also helps to develop a more generic and comprehensive picture of how different management practices influences N-oxide fluxes in the region. Inclusion of NO fluxes is especially exciting, because (as the authors note) we have little or no data on NO fluxes from oil palm systems so far. Given the importance of NO in regulating local/regional atmospheric chemistry (Fowler et al., 2011, Hewitt et al., 2009) and transport of reactive N across the landscape, understanding of NO fluxes could have more wide-ranging policy and management implications.

From an experimental perspective, the spatial sampling design was robust and the experiment was well-replicated. The authors are to be commended for collecting such a complete set of flux and environmental data from a remote field site with relatively poor infrastructure. The monthly sampling frequency is also adequate for capturing major trends in N-oxide fluxes, and the relationship between N-oxide fluxes and environmental variables. The higher frequency sampling study that investigated shorter-term trends in N-oxide flux following fertilisation also provides a good picture of how fertiliser application influences N-oxide fluxes. While it would be desirable to have fluxes collected at finer temporal resolution using quasi-continuous sampling methods (e.g. automated chambers), I do not believe that these kinds of data are necessary to test the questions and hypotheses posed here as the authors do not aim to construct an ecosystem N budget.

While I am strongly supportive of this work overall, I do have a few concerns. First, I

believe that the authors need to reconsider the structure of the Methods and Results sections to improve the clarity of the text. For the Methods section, I was sometimes confused as to which ecosystems/land-use were sampled at what times, and I think the authors should revise the sections describing the experimental design to better clarify the chronology of the measurements. From my reading of the text, it appears that there were 2 parts to this study; the first phase, where gas fluxes were compared among forest, jungle rubber, and small holder plantations. During the second phase, fluxes were compared among small holder and large holder plantations. It would be useful if the text could be edited to make this sampling design a bit clearer. In addition, measurements were discussed in the Results and Discussion which were not described in the Methods – for example, potential nitrification measurements were performed, but not described in the Methods. By inference, I had assumed that potential denitrification measurements had been conducted too, as the authors later conclude on Page 18, section 3.4 that nitrification was the dominant N-oxide producing process (which implies that other pathways such as denitrification or DNRA were not closely correlated with N-oxide fluxes). I had wondered if these potential nitrification measurements had been conducted as part of another study; if so, then this needs to be acknowledged.

Second, I thought that the structure of the Results section could be improved. I felt that the way in which the Results were organised did not convey information clearly about how fluxes varied among land-uses and soil types. In my opinion, I think it would be clearer if the first part of the Results compared trends among land-uses (e.g. forest, jungle rubber, small holders; small holders versus large holders, etc.). The authors could then go on to explore differences among soil types. The second part of the results section could discuss temporal trends in N-oxide fluxes, such as intra-annual trends in N-oxide fluxes (if any exist) as well as the pattern in N-oxide fluxes after fertilisation. The last part of the Results could discuss the role of environmental variables and N cycling processes (e.g. nitrification) in regulating flux rates. This could all be achieved without altering the text too much, but simply re-organising how the information is presented.

I had no major concerns about the Introduction and Discussion, as I felt that the authors did an excellent job of framing their research within a wider theoretical and applied context, and linking their findings back to bigger picture questions about the generic controls on N biogeochemistry in tropical soils.

Specific comments are provided in the section below.

SPECIFIC COMMENTS

1. Page 5, line 16-page 6, line 9: Generally, I think that this section describing the hypotheses and overall experimental goals is well-written. However, my concern here is how to introduce the second part of the study comparing N gas fluxes in small versus large holder systems in a more intuitive way. The current structure of this section makes the study on small versus large holder systems seem a bit disconnected from the first phase of the work. One possibility might be to introduce this study earlier on in the paragraph, close to the section where the authors pose their hypotheses (which implicitly refer to N availability and the HIP model), as this would then implicitly link-up to ideas about N control on N fluxes, e.g. (my suggestions in the underlined section below):

"We covered four different land uses within two landscapes on highly weathered soils that mainly differed in soil texture (clay and loam Acrisols): forest, rubber trees interspersed in secondary forest (hereafter called jungle rubber) as the reference land uses, and smallholder rubber and oil palm plantations as the converted land uses. In addition, we conducted a follow-on study comparing N gas fluxes across a gradient of N input that encompassed small holder plantations (lower N input rates) a large-scale oil palm plantations (higher N input rates) to try and evaluate the effect of N input rate on N gas fluxes..."

2. Page 7, lines 12-17: In the comparison study between small holder versus large holder systems, were measurements from the small holder systems collected at the same time (i.e. were fluxes from the two types of oil plantations collected concomitantly)? If so, then this should be made clearer in this paragraph.

3. Page 9, lines 7-17: It would be useful at the start of this paragraph to remind readers which land-uses were sampled in 2013, 2014 and 2015. Perhaps the authors could put together a table or something similar to represent this information?

4. Page 9, lines 18-20: Were the authors able to determine if N2O fluxes varied with distance from palms? Given the spatial structure in oil palm plantations, and the potential effects of roots and fertiliser application, it would be useful to know if the data could be corrected for spatial effects (if they exist) caused by proximity to palms.

5. Page 15, lines 16-25: I wonder if the large variation in the mean fluxes is driven by a high degree of within-plot spatial variability, which might linked to where fertiliser is applied, the distribution of palms, or surface residues (e.g. palm fronds or planted understory plants)? Is it possible to determine to what extent micro-scale variability, linked to spatial structure in the plantation, was causing variance in the measurements? This could help in interpreting the data, and understanding differences linked to management differences in small holder vs larger holder systems.

6. Page 16, lines 1-10: There is a potential confounding effect here due to the presence of roots which needs to be acknowledged. Granted, it is likely that the effect of fertiliser application will overwhelm the effect of roots in the immediate to short-term after fertilisation. However, it is worthwhile knowing whether or not the presence of roots ameliorates the effects of fertiliser (e.g. plant competition with nitrifiers/denitrifiers for inorganic N may reduce the relative gases loss of N in areas with high root densities). For example, do the authors have data on N gas fluxes from root-free and rhizosphere soil in the large holder systems to compare against? My thought here is that if the N application rate is higher in the large holder systems it may be possible to compare N fluxes from rhizosphere soil with different N application rates to evaluate the effect of N input rate on gas fluxes (i.e. making a like-for-like comparison).

7. Page 16, lines 1-17: Regarding the use of locations a, b and c to refer to different

distances to the palm; perhaps it may be possible to use identifiers that are a bit more descriptive, as this would make it easier for the readers to pick-up on the information quickly? e.g. 0.3 m = "inner root ball", 0.8 m = "outer root ball", 4-4.5 m = "inter-palm space" (or something similar)? Use of letters is a bit more abstract and (while clear) forces the reader to refer back to the tables or legends to remind themselves of the meaning of these abbreviations.

Also – where trends are statistically significant, the authors could list the P-values from the multiple comparisons tests in parentheses to highlight where significant trends existed (I see that this has been done for the table, but would be useful for the reader if this was stated in the text, too).

8. Page 16, lines 18-22: Are these estimates derived from the trapezoidal extrapolations or some form of area-weighted upscaling?

9. Page 18, section heading 3.3 Temporal controls of soil N-oxide fluxes: This section appears to discuss the relationship between environmental variables/drivers and N-gas fluxes. Perhaps it may be more appropriate to re-name this section as "Role of abiotic variables in controlling N-oxide fluxes"? Or, if the authors may wish to more explicitly discuss how temporal variability in these environmental drivers contribute to fluctuations in N-oxide fluxes?

10. Page 18, section heading 3.4 Spatial controls of annual soil N2O fluxes: Similar to my above point (9), I do not feel that this heading properly describes what is discussed in the section. In this section, the authors discuss the relationship between N cycling processes rates and N-oxide fluxes, in order to evaluate the principal source of N-oxides in these soils. They conclude that nitrification is probably the dominant driver of N-oxide fluxes because of the correlation between nitrification rates and gas fluxes. Perhaps the section could be retitled "Role of different N cycling processes in regulating N-oxide fluxes"?

Also – I re-read the Methods and did not see the nitrification potential experiments

described. Was this work done as part of another study or was this done as part of this work? In either case, this needs to be added to the Methods to make it clear that this work was done as the reference to nitrification (although interesting and relevant) came as a but of a surprise.

11. Page 20, lines 9-22: Fluxes of NO from these systems, particularly oil palm, is extremely novel and of wider environmental significance, given the potential role of NO in tropospheric ozone formation, N deposition, and regional atmospheric oxidant (OH) balance. It would be useful in the discussion if the authors could bring into the discussion some of the findings from earlier atmospheric sampling campaigns by the OP3 consortium (Fowler et al., 2011, Hewitt et al., 2009), where elevated NOx concentrations were found in the troposphere near oil palm plantations? Hewitt et al. (2009) and Fowler et al. (2011) suggest that the implications of enhanced NO emissions from oil palm could be potentially regionally significant, and the work here in Sumatra on ground-based NO fluxes would be an interesting counter-point to the atmospheric sampling work from Sabah.

REFERENCES

FOWLER, D., NEMITZ, E., MISZTAL, P., DI MARCO, C., SKIBA, U., RYDER, J., HELFTER, C., CAPE, J. N., OWEN, S., DORSEY, J., GALLAGHER, M. W., COYLE, M., PHILLIPS, G., DAVISON, B., LANGFORD, B., MACKENZIE, R., MULLER, J., SIONG, J., DARI-SALISBURGO, C., DI CARLO, P., ARUFFO, E., GIAMMARIA, F., PYLE, J. A. & HEWITT, C. N. 2011. Effects of land use on surface–atmosphere exchanges of trace gases and energy in Borneo: comparing fluxes over oil palm plantations and a rainforest. Philosophical Transactions of the Royal Society of London B: Biological Sciences, 366, 3196-3209.

HEWITT, C. N., MACKENZIE, A. R., DI CARLO, P., DI MARCO, C. F., DORSEY, J. R., EVANS, M., FOWLER, D., GALLAGHER, M. W., HOPKINS, J. R., JONES, C. E., LANGFORD, B., LEE, J. D., LEWIS, A. C., LIM, S. F., MCQUAID, J., MISZTAL, P.,

MOLLER, S. J., MONKS, P. S., NEMITZ, E., ORAM, D. E., OWEN, S. M., PHILLIPS, G. J., PUGH, T. A. M., PYLE, J. A., REEVES, C. E., RYDER, J., SIONG, J., SKIBA, U. & STEWART, D. J. 2009. Nitrogen management is essential to prevent tropical oil palm plantations from causing ground-level ozone pollution. Proceedings of the National Academy of Sciences, 106, 18447-18451.

---

## Author Comment (AC1) · 1 Feb 2017

We would like to thank Referee #1 for the time he/she invested to give his/her thoughtful and constructive comments. For clarity, we have copied his/her comments and placed our answers below each comment.

GENERAL COMMENTS

Referee: A general problem I have with this manuscript is the length of many sections and the wordiness of many paragraphs. The abstract alone comes in at ∼430 words and could be substantially shortened (no need to describe site replication for instance). The "Materials and Methods" section for instance is extremely long (9.5 pages) and should be streamlined.

Answer: In order to address the referee's concerns, we shortened the abstract. As suggested we took out the replication and the timing of sampling (page 2, lines 10-15 of the original manuscript and removed the sentences about N-oxide losses of the applied N in the abstract (page 3, lines 1-3 of the original manuscript). We also streamlined section 2.1 "Study area, experimental design and management practices". We took out details regarding the management practices, which have been reported in our earlier paper (Hassler et al., 2015) and which have been not directly relevant in the present manuscript (page 8, lines 6-15 of the original manuscript). Lastly, we shortened section 2.3 "Statistical analysis" (see answer to detailed comment #13 below) and put the detailed statistical description to Appendix A.

Referee: In contrast, the discussion in particular would benefit from greater detail (and discussion with results from other regions of the world).

Answer: We indeed compared our results on soil N-oxide fluxes with measurements from other parts of the world (page 19, lines 7-8; pages 19-20, lines 22-2; page 21-22, lines 22-2; page 22, lines 13-16). This we elaborated also in our answer to comment #27 below).

Referee: Also to me, a clear site/ replicate nomenclature would better guide the reader through the text as the full measurement setup is rather complex (two soil landscapes, 4 land uses, 3 chamber positions at each site, 4 replicates). For instance, if the authors could define some site abbreviations (e.g., reference land uses: F (forest), JR (jungle rubber); converted uses: RP (rubber plantation), PP (oil palm plantation), they could simple use to those instead to repeat the site attributes or be overly descriptive. This is also true for the naming of the three within-site chamber positions (currently: a, b, c). While their properties are described in the text, and also in the caption of table 4 it makes the digestion of the data presented unnecessary hard for the reader (maybe: F1 (fertilised area position 1 / 0.3m from stem), F2 (fertilised area position 1 (0.8m from stem), NF (non fertilised: 4.5m from stem). In lengthy paragraphs it is easy to get lost and scramble to read up what i.e. position b represents (same for the reference to the proposed hypothesis').

Answer: We agree with the suggestion to use more descriptive abbreviations for chamber positions a, b and c. To address this concern, we introduced the abbreviations: F1 = chamber at 0.3 m from the tree base with incidental fertilization, F2 = fertilized chamber at 0.8 m from the tree base, NF = non-fertilized chamber location at 4-4.5 m from the tree base (page 10, lines 2-6 and Tables 2 & 4). We also, as suggested, included abbreviations for the hypotheses (H1 and H2) (page 5, lines 19-24) and pointed to these abbreviated hypotheses in the discussion to remind the reader how we linked our findings with the hypotheses (e.g., page 13, lines 15-18). On the other hand, we did not use abbreviations for the land uses in order to avoid confusions with all the abbreviations.

Referee: Furthermore, I feel that reorganising and cleaning the tables would help the better digest the main results presented. Table A1 & A2 should be combined and added to the main text. The figures are appropriate, but could also be improved, too (see detailed comments).

Answer: Please find our answers regarding this comment below (detailed response).

DETAILED RESPONSE:

Referee: In the following I'd like to suggest some changes to the tables (often admittedly personal preference)

Table 1.

- Referee: Shorten the caption (16 lines of description).

Answer: The reason why the table titles are long is because in Biogeosciences, the table format must not have a footnote. This table title would have been short if the parts on statistical tests and identifiers can be placed as a footnote. We shortened the table titles (Tables 1-4) by taking out details on the measurement period, which are now put in Appendix Table A1 (as suggested by reviewer 2). The table title, however, has to be succinctly correct without too much reference to the text, as one of the criteria of a Table is that it has to be completely understandable without referencing too much to the main manuscript.

- Referee: Also a column with number of samples (n) would help the reader to assess the robustness of the given average emission.

- Answer: We included this information directly after the SE on the first line of the table title in order to minimize columns with anyway the same entry for n = number of 'real' replicate sites or plots per land use.

- Referee: I would suggest to round to the first decimal to reduce visual clutter (esp. with the group identifiers presented in the table)

- Answer: We agree with the suggestion and reduced the decimal place to one.

Table 2

- Referee: The chamber location identifiers a, b and c do not help the reader. Either also identify the distance to the tree in the table or use descriptive abbreviations

- Answer: As the reviewer suggests in the general comments #3 above, we introduced the abbreviations: F1 = chamber at 0.3 m from the tree base with incidental fertilization, F2 = fertilized chamber at 0.8 m from the tree base, NF = non-fertilized chamber location at 4-4.5 m from the tree base.

- Referee: Again, indicate the number of measurements considered

Answer: The number of measurements is already indicated in the table title (line 3), and also the number of replicates is given in table title (line 1).

- Referee: Given the lack of NO data for 4 of the 6 sampled sites, maybe another organization would be better. For instance: Table 2a (N2O) – columns: Oil palm site /

ch pos / N2O (clay Acrisol) / N2O (loam Acrisol)

Table 2b (NO)- columns (with oil palm sites given as NO column identifier (?): CH pos / NO (clay Acrisol) / NO loam Acrisol

- Answer: Although this is generally a very good suggestion, we are convinced that the given structure of the table is also reasonable because of the following reasons: At a glance it is possible to see the differences in NO and N2O fluxes for the sites where both gases were measured. The structure follows that of Table 1, to which the reader can easily cross-reference. To reduce clutter and improve ease in reading, the values are rounded off to one decimal place.

Table 3./4.

- Referee: They could go into the appendix

- Answer: We put all supplementary information now in Appendix A and Appendix Tables A1-A3. We decided to keep Tables 3 and 4 in the main manuscript because one of our objectives is to determine the controlling factors of soil N-oxide fluxes. These tables show the important controlling factors in the different land uses and also following fertilization.

- Referee: Table 4 should be split into N2O and NO data (see Table 2)

- Answer: Splitting Table 4 into two for these two gases will increase unnecessarily the number of tables, when in fact N2O and NO can simply be put in the same table and it is easy to read this table for these two gases. This Table has similar structure as Table 3 in our earlier study (Hassler et al. 2015), published in Biogeosciences, where the two gases, CO2 and CH4, are put in the same table with their soil controlling factors.

Table A1/A2.

- Referee: Shorten caption

- Answer: In case of Table A1 (now Table A2 in the revised manuscript), we addressed this concern by taking out the information on the statistical analysis because the significant differences among land uses and between landscapes do not play a role on how we discussed the influence of these parameters on soil N-oxide fluxes. - In case of Table A2 (now Table A3 in the revised manuscript), we addressed this concern by taking out the information about their measurement period, since this was also described in the methods (page 12, lines 10-13). We kept the statistics in this case because we refer to differences in soil mineral N content in our discussion (page 22, lines 17-21).

- Referee: Combine A1 and A2 into one table and add it into main document as a site description/ reference for the reader.

- Answer: We are convinced that these tables should stay separated for clarity reasons: First, the data in Table A1 (now Table A2) are used for the determination of spatial controls on annual soil N2O fluxes and are only determined once, while the data in Table A2 (now Table A3) were determined concurrently with the soil N-oxide flux measurement (page 12, lines 10-13) and are used for determination of temporal controls on monthly measured soil N2O fluxes (section 3.3). Second, a general site description is given in section 2.1, and Table A2 not aimed to describe the site but to give supporting data, which show correlations with annual soil N2O fluxes.

- Referee: Round WFPS, NH4 and NO3 to one digit to reduce clutter

- Answer: We rounded the values to one decimal place.

- Referee: This might be personal preference, but maybe remove the significance letters, too (they make the table really hard to read, also almost all entries in A1 have a lowercase 'a', maybe only label when they differ?; and important differences can be discussed in the manuscript).

- Answer: We agree with this suggestion and removed the statistical analysis from Table A1 (now Table A2).

Figures.

Referee: Some scale modification and additional labels would make the figures easier to read.

Fig. 2. - Referee: Matching scales would help the reader (at least 4 groups; a) & c) and b) & d)

- Answer: We tried this suggestion but it did not improve clarity. Instead, it diminished temporal pattern of the fluxes following fertilization. The reason is because soil N2O fluxes at F1 (0.3 m from the tree base with incidental fertilization; Figs. 2a and 2b) would be so minimized because of its much lower fluxes than those at F2 (fertilized location; Fig. 2c and 2d). Thus, we kept the original figure.

- Referee: Add Tree-base distance in the plots to guide the reader

- Answer: We included these now on the figure panels, in addition to the fact that they were actually included in the figure caption.

- Referee: Add fertilizer amounts to plot or captions (instead of referring section 2.2)

- Answer: We included the amount of added N to the caption.

Fig. 3

- Referee: See comments for figure 2 (y-axis break for a) and b) required)

- Answer: Please see author's answers for Figure 2 above.

DETAILED COMMENTS

1) Referee: p5, l19: Introduce site abbreviations that you can refer to in the text

Answer: We believe that introducing site abbreviations for the four land uses will not improve clarity but could confuse the reader, since we also used abbreviations for the different chamber locations within the smallholder oil palm plantations (see answer to general comment #3 above).

2) Referee: p5, l22: Introduce H1 and H2 for your hypothesis so you can refer to them in your discussion

Answer: We included this suggestion (see answer to general comment #3).

3) Referee: p6: I would give a site property table here (basically combine A1&A2) and add soil properties – I feel a reference to Allen et 2015 and Hassler 2015 for such fundamental information for the manuscript is not sufficient.

Answer: We gave all the necessary site characteristics, including soil characteristics (page 6, lines 19-24), in section 2.1. Therefore, an additional table for site general characteristics is not necessary (see also our detailed response to comment on Table A1/A2 above).

4) Referee: p6, l16: is the precip data given as SD?

Answer: The precipitation data are the mean of the years 1991-2011 with the standard error among these measurement years.

5) Referee: p6, l20: that is actually substantially higher

Answer: That is true and therefore we highlighted this fact.

6) Referee: p7-8: site & design description could be shortened substantially

Answer: We streamlined section 2.1 by taking out details, mainly regarding the management practices (see answer to general comment #1 above).

7) Referee: p8: Please work on the language in this section: I counted 'was done', 5 times in this paragraph

Answer: After removing lines 6-15, we hope this language shortcomings are also remedied.

8) Referee: p9, l20: give a reference for N fertilizer induced pulse emissions

Answer: We cited Veldkamp and Keller (1997) and Veldkamp et al. (1998) who reported fertilizer-induced pulse emissions (page 9, lines 10-13).

9) Referee: p11: trapezoidal rule should be explained briefly here (esp. since it is not explained in the given reference Hassler at al., 2015, either.; in there is an other reference to Koehler at./ Veldkamp 2013).

Answer: For clarity reasons, we rephrased this sentence. Basically, trapezoidal rule is the simple interpolation between measured fluxes and the interval between sampling days (page 11, line 14-17).

10) Referee: p12, l4-l10: this is hard to read; just give the equation

Answer: We rewrote these lines into an equation form, aligned to the left margin for ease in reading (page 12, lines 1-4).

11) Referee: p13, l10: "when necessary" – explain

Answer: We improved the sentence; we meant, when assumptions for normal distribution and homogeneity of variance were not met (page 13, lines 13-15).

12) Referee: p13, l13: briefly remember your reader about your hypothesis H1 & H2 here

Answer: We added the hypotheses into the brackets of the sentence (page 13, lines 15-18).

13) Referee: p13, l22- p14, l15: this is very detailed... maybe move this into the appendix/ a supplement?

Answer: We agree with the referee's suggestion and put the detailed statistical description regarding the use of LME models to Appendix A. We revised section 2.3 "Statistical analysis" to describe which comparisons these LME models were applied (page 13, lines 15-24).

14) Referee: p15, l11: mention the reference land uses again

Answer: We included them in brackets (page 15, lines 1 and 7).

15) Referee: p15, l11: "...from soils. In the clay..."

Answer: We take this suggestion (page 15, line 1-2).

16) Referee: p15, l15: Was this systematic? I.e., was there always one measurement (position) an outlier?

Answer: This was not systematic. This occurred in all land uses, where one or two plots in some sampling days displayed higher emissions than the other plots of the same land use within the same landscape.

17) Referee: p16, l3-4: give the fertilizer rates here, too.

Answer: We included them into the brackets (page 15, line 23).

18) Referee: p16, l6: "in the chamber location closest to the tree, soil N2O emissions..."

Answer: We included this suggestion to make clearer which location we are talking about (pages 15-16, lines 25-2).

19) Referee: p16, l9: There is also a peak for site 1 (but smaller)

Answer: That is true! Nevertheless, mean fluxes were statistically not different between chamber location F1 and NF in this site, and hence we only highlighted site 2.

20) Referee: p16, l18: Due to which assumptions? Trees per ha? Avg. basal area of those trees?

Answer: The area-coverage calculation of fertilizer-induced N-oxide emissions was based on the number of trees/hectare. We made this clearer by including this information into the brackets (page 16, lines 13-14).

21) Referee: p18, l3: NH4 (only weak?)

Answer: For NH4+, only a weak correlation was found. We stated in section 2.3 (page 14, lines 19-20) that correlations with marginal significant will be included, and this is also clearly identified in Table 3.

22) Referee: p18, l5: What is the temperature amplitude between the measurements? Relatively minor I suppose due to the tropical climate.

Answer: The soil temperatures of the sampling days during 1 yr of measurement in the converted land uses ranged between 24.4 °C and 30.6 °C. This correlation with soil temperature was definitely because of fertilizer-induced high N2O emissions on one sampling day with relatively high soil temperatures (28.8° C).

23) Referee: p18, l5-9: Remove this single sampling period outright since it clearly seems fertilizer-induced

Answer: We removed these lines and revised Table 3, according to this suggestion, since we also believe that this information is unnecessary.

24) Referee: p18, l13-14: How is this possible?

Answer: NO fluxes following fertilization in chamber location F2 (formerly, chamber b) of the clay Acrisol did not correlate with mineral N contents but instead correlated negatively with WFPS. The clay soil had high water retention capacity and WFPS over-shadowed the influenced of mineral N on soil NO fluxes (in a condition with sufficient mineral N availability from fertilization) – soil NO fluxes were favored under conditions of low WFPS. Conversely, in the loam Acrisol, where WFPS were lower than those in the clay Acrisol, and favored for soil NO fluxes, NO was more influenced by mineral N than by WFPS. This is exactly what we discussed in pages 25-26, lines 20-2.

25) Referee: p19, l17: Give the range of your fluxes here for comparison

Answer: The reason why we don't write in the text the fluxes reported clearly in Table 1 is to avoid redundancy. One clear requirement is that values reported in Tables should never be repeated in the text; instead, we referred to the Table where these values are reflected.

26) Referee: p19, l19-25: This is very wordy, could be shortened substantially

Answer: We need to provide in these sentences the frequency of sampling and spatial replications of the studies to which we compared our values. These are very much needed in order to understand why fluxes from other studies are higher or lower compared to our measured fluxes. Still, we understand the referee's concern and therefore we removed the information on elevation (since we state anyway if we are talking about lowland forests or montane forests) and removed the decimal places but retain the sampling frequency and spatial replication (page 19, lines 8-18).

27) Referee: p20, l8: What about the other literature? You only compare to reports from your specific region

Answer: The reviewer is referring here only the summary statement of our comparisons with other values. We indeed relate our measured fluxes not only with previous studies within Indonesia but also with those studies in other tropical regions. In the first part of this paragraph, we indeed compared our soil N2O fluxes with values reported in literature (page 19, lines 7-8) and we also put these findings in a broader context (page 20, lines 7-10).

28) Referee: p21, l24: I do not get the reasoning here. Were the fires going on in the region during the measurements?

Answer: Fires are regularly occurring in Jambi region and in the whole of Sumatra Island. During fires, NO levels are generally elevated (Levine, 1999). To highlight this, we included Gaveau et al. (2014), who reported Sumatran fires 2013 (page 21, line 13-14).

29) Referee: p22, l8: Give the observed flux range here for better comparison, also the N application rates would help to judge the observations.

Answer: We did not repeat putting in our measured fluxes in the text as these are clearly stated in Table 1 to which we referred. To minimize the wordiness of our com-

parison with other reported studies, we removed the information on elevation, but retained the information on replication and sampling frequency. We agree that stating N fertilization rates is important, and we included them if they were provided by the cited studies (pages 21-22, lines 22-16).

30) Referee: p22, l9: Why do you give the elevation here? This is not really a factor (110m, 580,...)

Answer: We took out this aspect (see answer to detailed comment #26).

31) Referee: p22, l12: However the sampling there was very detailed and covered the transition period

Answer: We provided uniformly for all cited studies the frequency of measurement or the duration of measurement, whichever is given by the cited studies, and the replications so that the readers have the full background on how to judge the differences in flux values reported by these studies (see answer to detailed comment #26).

32) Referee: p22, l15: "nine monthly" is a bit deceptive, it's 9 single measurements, right?

Answer: Nine sampling days at monthly interval.

33) Referee: p22: Maybe a literature review table with relevant citations for the investigated land uses combined with your results would be appropriate here? This would also help the better interpret your results in context.

Answer: Yet another table will not shorten this manuscript. Besides there are only few studies that measured soil N2O fluxes (nothing for soil NO fluxes) from the same land uses to warrant another table. We are convinced that by having incorporated the above mentioned changes (removing information on elevation and rounding off values to have no decimal place) improve this section.

34) Referee: p23, l4-l6: Also true, this seems unnecessary to mention here. Maybe give a half-sentence in the abstract highlighting the novelty of your NO measurements.

Answer: We agree with this comment and have removed this sentence, since we also mention this aspect in the conclusion (page 26, lines 12-14).

35) Referee: p23, l7: remind the reader about the hypothesis again

Answer: We referred back to the hypothesis number, but not rewriting it again in order to avoid redundancy. The reader can easily get back to the hypotheses now that these are referred to in numbers (page 22, lines 17).

36) Referee: p24, l21: Isn't it expected that fertilizer-induced emissions occur at the site where fertilizer is applied?!?

Answer: Yes, of course. But we want to point out here that for banded (meaning around a small area from the tree base) fertilizer application, as was claimed to be practiced by our smallholders, fertilizer-induced N-oxide emissions are limited within the fertilized area and only lasted within a short time. This may be different for large-scale plantations where fertilizers are broadcasted in much larger amounts.

37) Referee: p25, l8: mention your fertilizer rates again for comparison

Answer: We included the rates, as suggested (page 24, line 24).

38) Referee: p25, l9: these seem high; please give the references

Answer: We included the references again: Veldkamp and Keller, 1997; Veldkamp et al., 1998 (pages 24-25, lines 25-1).

39) Referee: p25, l25: pulse application? Maybe: "the event-based application of high N rates" or something similar?

Answer: We agree that this is an awkward wording, and deleted the word "pulse" (page 25, line 17).

40) Referee: p26, l10-12: This is most likely not true for low – medium moisture levels.

Answer: The referee points out that soil NO fluxes do not decrease and soil N2O fluxes do not increase under low to medium moisture levels. According to Davidson et al. (2000) soil N2O fluxes start to increase at around 30 % WFPS and soil NO fluxes start to decrease at around 60 % WFPS. WFPS in site 1 of the loam Acrisol soil (chamber locations F1 and F2, which showed a positive correlation between soil N2O fluxes and WFPS) ranged between 25 and 45 %. WFPS in in site 3 of the clay Acrisol (chamber location F2, which showed a negative correlation between soil NO fluxes and WFPS) ranged between 46 and 68 %. The correlation between N-oxide fluxes and WFPS follows the expected pattern based on the HIP model. Therefore, we are convinced that this sentence is correct.

41) Referee: p26, l12-16: This sentence actually highlights a key problem with such extensive sampling routine and should be discussed further.

Answer: We believed we have indeed discussed this extensively by pointing out the temporal patterns following fertilization at each smallholder site. Besides, an extensive sampling is not a problem but indeed a solution to include the short-term effect of fertilization. It is clear from our results that soil N-oxide measurements should be accompanied with concurrent measurements of known controlling factors; otherwise, investigators will not be able to explain their results. We are indeed able to explain this temporal pattern because we have recognized the simultaneously decreasing WFPS and increasing mineral N content over time in this site 3 of the loam Acrisol soil.

42) Referee: p26, l20: true, although the "full year" is based on few measurements

Answer: We also do not want to overrate our study and therefore clearly stated again, that our measurements were conducted on a monthly basis (page 26, lines 12-13).

43) Referee: p26, l22: Name the hypothesis, the reader might have forgotten which hypothesis was which

Answer: We referred to the hypothesis number (also see answers above for the same comments).

44) Referee: p27, l7: ditto

Answer: Please see answer above.

45) Referee: p27, l12: change unit 'kg' to 't'

Answer: We changed "kg" to "tons" (page 27, line 4).

**Supplement:**

[revised manuscript text omitted]

| Oil palm (smallholder plantation) | 12.2 ± 6.1[a,A] | 0.7 ± 0.7[ab,A] | 1.1 ± 0.5
*0.1 ± 0.0* |
| Oil palm (large-scale plantation) | 42.3 ± 24.2[a,A] | - | 3.3 ± 1.7 |

**Table 2.** Mean (±SE, $n = 3$ oil palm trees) soil $N_2O$ and NO fluxes from three chamber locations during a fertilization in three (for $N_2O$) or one (for NO) smallholder oil palm plantation within each landscape, measured 6 to 11 times during 3–8.5 weeks following fertilization. Means followed by different letters indicate significant differences among chamber locations within each site (linear mixed-effect models with Fisher's LSD test at $P \leq 0.05$). Chamber F1, F2 and NF were placed at 0.3 m (with incidental fertilization), 0.8 m (fertilized area), and 4–4.5 m (non-fertilized area, serving as the reference chamber), respectively, from the stem base. 0.32 kg N tree$^{-1}$ was applied in the clay Acrisol and 0.26 kg N tree$^{-1}$ in the loam Acrisol in accordance to the smallholders' practices.

| Oil palm site | Chamber location | $N_2O$ fluxes ($\mu g\ N\ m^{-2}\ h^{-1}$) | NO fluxes ($\mu g\ N\ m^{-2}\ h^{-1}$) |
|---|---|---|---|
| clay Acrisol landscape | | | |
| 1 | F1 | $156.7 \pm 86.8^b$ | - |
| | F2 | $910.1 \pm 410.0^a$ | - |
| | NF | $6.9 \pm 3.3^c$ | - |
| 2 | F1 | $130.6 \pm 34.6^b$ | - |
| | F2 | $692.7 \pm 144.1^a$ | - |
| | NF | $9.9 \pm 3.0^c$ | - |
| 3 | F1 | $45.5 \pm 3.7^b$ | $4.7 \pm 1.7^b$ |
| | F2 | $1281.0 \pm 486.7^a$ | $535.3 \pm 194.5^a$ |
| | NF | $1.1 \pm 1.6^c$ | $1.5 \pm 1.5^b$ |
| Oil palm site | Chamber location | $N_2O$ fluxes ($\mu g\ N\ m^{-2}\ h^{-1}$) | NO fluxes ($\mu g\ N\ m^{-2}\ h^{-1}$) |
| loam Acrisol landscape | | | |

| | | | |
|---|---|---|---|
| 1 | F1 | $33.5 \pm 9.8^{b}$ | - |
| | F2 | $133.4 \pm 34.9^{a}$ | - |
| | NF | $11.8 \pm 6.1^{b}$ | - |
| | | | |
| 2 | F1 | $129.7 \pm 46.2^{a}$ | $46.2 \pm 19.6^{b}$ |
| | F2 | $205.3 \pm 24.2^{a}$ | $157.1 \pm 35.7^{a}$ |
| | NF | $7.9 \pm 4.8^{b}$ | $0.7 \pm 0.3^{b}$ |
| | | | |
| 3 | F1 | $5.2 \pm 1.0^{b}$ | - |
| | F2 | $104.5 \pm 81.9^{a}$ | - |
| | NF | $3.7 \pm 1.7^{b}$ | - |

**Table 3.** Pearson correlation coefficients between soil $N_2O$ flux ($n$ = 48; µg N m$^{-2}$ h$^{-1}$), soil

NO flux ($n$ = 16; µg N m$^{-2}$ h$^{-1}$), water-filled pore space (WFPS; %, top 0.05 m depth), soil temperature (°C, top 0.05 m depth) and extractable mineral N (mg N kg$^{-1}$, top 0.05 m depth)

across landscapes for the reference and converted land uses. Correlation was conducted using the means of the four replicate plots per land use on each of the 12 monthly measurements (for soil $N_2O$ fluxes) and four monthly-bimonthly measurements (for soil NO fluxes).

| Land-use type | Variable | WFPS | Soil temp. | $NH_4^+$ | $NO_3^-$ |
|---|---|---|---|---|---|
| Reference land uses (forest and jungle rubber) | Soil $N_2O$ flux | -0.21 | -0.09 | -0.23 | 0.38[c] |
| | Soil NO flux | -0.74[c] | -0.15 | -0.48[a] | 0.69[c] |
| Converted land uses (rubber and oil palm) | Soil $N_2O$ flux | 0.11 | 0.15 | 0.23 | 0.37[c] |
| | Soil NO flux | -0.05 | 0.09 | -0.05 | 0.23 |

[a]$P \leq 0.09$, [b]$P \leq 0.05$, [c]$P \leq 0.01$.

**Table 4.** Pearson correlation coefficients ($n$ = 6–11 measurements following fertilization)

between N-oxide fluxes ($\mu$g N m$^{-2}$ h$^{-1}$), water-filled pore space (WFPS; %, top 0.05m depth)

and extractable mineral N (mg N kg$^{-1}$, top 0.05 m depth), measured at different chamber locations (F1, F2 and NF were at 0.3 m (with incidental fertilization), 0.8 m (fertilized area)

and 4–4.5 m (non-fertilized area), respectively, from stem base). Correlation was conducted using the means of the three replicate trees per chamber location. 0.32 kg N tree$^{-1}$ was applied in the clay Acrisol and 0.26 kg N tree$^{-1}$ in the loam Acrisol in accordance to the smallholders'

practices.

| Oil palm plantation site | Chamber location | Variable | WFPS | $NH_4^+$ | $NO_3^-$ |
|---|---|---|---|---|---|
| clay Acrisol landscape | | | | | |
| 1 ($n$ = 6 measurements) | F1 | Soil $N_2O$ flux | 0.55 | 0.88[b] | 0.46 |
| | F2 | | 0.57 | -0.22 | -0.31 |
| | NF | | 0.37 | -0.64 | -0.44 |
| 2 ($n$ = 11 measurements) | F1 | Soil $N_2O$ flux | 0.11 | 0.93[c] | 0.95[c] |
| | F2 | | 0.08 | 0.05 | -0.06 |
| | NF | | 0.09 | -0.44 | -0.45 |
| 3 ($n$ = 10 measurements) | F1 | Soil $N_2O$ flux | -0.19 | 0.10 | 0.09 |
| | F2 | | 0.05 | 0.86[c] | 0.85[c] |
| | NF | | -0.32 | 0.06 | -0.44 |
| 3 ($n$ = 10 measurements) | F1 | Soil NO flux | -0.34 | 0.44 | 0.48 |
| | F2 | | -0.61[a] | 0.10 | -0.04 |
| | NF | | 0.59[a] | -0.14 | -0.13 |
| loam Acrisol landscape | | | | | |
| 1 ($n$ = 6 measurements) | F1 | Soil $N_2O$ flux | 0.96[c] | -0.18 | 0.03 |
| | F2 | | 0.78[a] | 0.61 | -0.40 |
| | NF | | -0.06 | -0.29 | <0.01 |

| | | | | | |
|---|---|---|---|---|---|
| 2 (n = 9 measurements) | F1 | Soil $N_2O$ flux | -0.55 | $0.71^b$ | -0.03 |
| | F2 | | 0.35 | -0.20 | $0.89^c$ |
| | NF | | 0.34 | <0.01 | -0.35 |
| 3 (n = 11 measurements) | F1 | Soil $N_2O$ flux | $-0.68^b$ | $0.67^b$ | $0.62^b$ |
| | F2 | | -0.27 | -0.2 | $0.57^a$ |
| | NF | | 0.36 | 0.19 | 0.06 |
| 2 (n = 9 measurements) | F1 | Soil NO flux | -0.07 | 0.18 | -0.27 |
| | F2 | | 0.07 | -0.11 | $0.96^c$ |
| | NF | | -0.16 | 0.12 | -0.23 |

$^aP \leq 0.09$, $^bP \leq 0.05$, $^cP \leq 0.01$.

[revised manuscript text omitted]

---

## Author Comment (AC2) · 1 Feb 2017

We thank Dr. Yit Arn Teh for the time he invested to give his thoughtful and constructive comments. For clarity, we have copied his comments and placed our answers below each comment.

GENERAL COMMENTS

Referee: While I am strongly supportive of this work overall, I do have a few concerns. First, I believe that the authors need to reconsider the structure of the Methods and Results sections to improve the clarity of the text. For the Methods section, I was sometimes confused as to which ecosystems/land-use were sampled at what times,

and I think the authors should revise the sections describing the experimental design to better clarify the chronology of the measurements. From my reading of the text, it appears that there were 2 parts to this study; the first phase, where gas fluxes were compared among forest, jungle rubber, and small holder plantations. During the second phase, fluxes were compared among small holder and large holder plantations. It would be useful if the text could be edited to make this sampling design a bit clearer.

Answer: We addressed this concern by introducing earlier on (in the abstract, in section 1 "Introduction", 2.1 "Study area and experimental design" and in section 2.2 "Soil N-oxide fluxes and supporting soil factors") the study coverage, as suggested by Dr. Teh. We now stated in the revised manuscript that there are two parts of the study; the first was on quantifying soil N-oxide fluxes from the four different land-uses (page 6, lines 24-25; page 8, line 22; page 9, line 10), and the second, as a follow-on study, was on the comparison between smallholder and a large-scale oil palm plantations in the loam Acrisol soil (page 2, lines 10-12; page 5, lines 16-19, page 7, lines 9-12; page 9, lines 5-9).

Referee: In addition, measurements were discussed in the Results and Discussion which were not described in the Methods – for example, potential nitrification measurements were performed, but not described in the Methods. By inference, I had assumed that potential denitrification measurements had been conducted too, as the authors later conclude on Page 18, section 3.4 that nitrification was the dominant N-oxide producing process (which implies that other pathways such as denitrification or DNRA were not closely correlated with N-oxide fluxes). I had wondered if these potential nitrification measurements had been conducted as part of another study; if so, then this needs to be acknowledged.

Answer: We mentioned in section 2.3 "Statistical analysis" (page 14, lines 13-15) that we assessed the spatial control of soil biochemical characteristics on annual soil $N_2O$ fluxes, using the soil biochemical data reported in Appendix Table A2; in this table we reported the source of these data. To improve clarity in our present manuscript, we now mentioned in section 2.2 "Soil N-oxide fluxes and supporting soil factors" (page 13, lines 4-8) and section 3.4 "Spatial controls of annual soil N2O fluxes" (page 18, lines 20-22) the source of these soil biochemical data. We mentioned that these data were reported earlier by Allen et al. (2015) and in our present manuscript we only put in Appendix Table A2 the parameters that showed significant relationships with the annual soil N2O fluxes. Furthermore, the entire internal soil-N cycling was quantified in situ (except for denitrification) by Allen et al. (2015), and we used all the parameters of the soil-N cycling to correlate with the annual soil N2O fluxes. Only the gross nitrification rates showed significantly correlation with annual N2O fluxes from the reference land uses across the two landscapes. We did not interpret this correlation as the responsible process for N2O emission/production in the soil. Instead, we interpreted this as the control of soil N availability on soil N2O emission/production and gross nitrification as an index of soil N availability. Quantifying the relative importance of nitrification and denitrification on soil N2O fluxes from these land uses (which cannot be drawn from our data via mere correlation test) is the focus of a follow-on study by our group during the 2nd phase of this project, which has just started this year, 2017.

Referee: Second, I thought that the structure of the Results section could be improved. I felt that the way in which the Results were organised did not convey information clearly about how fluxes varied among land-uses and soil types. In my opinion, I think it would be clearer if the first part of the Results compared trends among land-uses (e.g. forest, jungle rubber, small holders; small holders versus large holders, etc.). The authors could then go on to explore differences among soil types. The second part of the results section could discuss temporal trends in N-oxide fluxes, such as intraannual trends in N-oxide fluxes (if any exist) as well as the pattern in N-oxide fluxes after fertilisation. The last part of the Results could discuss the role of environmental variables and N cycling processes (e.g. nitrification) in regulating flux rates. This could all be achieved without altering the text too much, but simply re-organising how the information is presented.

Answer: Although we understand why the reviewer is suggesting this sequence of flow in the Results, we ask for Dr. Teh's consideration of the basis of our decision on how we had organized the present structure of our results. The main reason of organizing the results this way (N-oxide fluxes 1st from the reference land uses with comparison between the two landscapes and followed by the converted land uses within each landscape, including the smallholder and large-scale oil palm plantations within one landscape, then fertilization effects as the most important management in oil palm plantations, and finally the temporal and spatial controls) is because we need first to establish if from the reference land uses (with no to minimal human disturbance) there are differences between the two landscapes, as baseline data for soil N-oxide fluxes, before going onto the land-use change effect and further onto the controlling factors. This flow of the result presentation also supports the sequence of logic in the discussion section:

a) how our measured fluxes from the baseline reference land uses are comparing with the other findings in the tropics and, with that, establishing how the soil factors control the temporal and spatial patterns of soil N-oxide fluxes.

b) how land-use conversion affected these fluxes and changed the controlling factors, and hence why significant change in N-oxide fluxes was not detected among land uses.

c) finally, the effects of fertilization, and that its importance for improved estimates of annual N-oxide fluxes at a scale larger than our present study lies on the inclusion of large-scale, more intensively fertilized oil palm plantations.

Referee: I had no major concerns about the Introduction and Discussion, as I felt that the authors did an excellent job of framing their research within a wider theoretical and applied context, and linking their findings back to bigger picture questions about the generic controls on N biogeochemistry in tropical soils.

Specific comments on individual portions of the text are provided in the section below.

[Figure]

SPECIFIC COMMENTS

1. Referee: Page 5, line 16-page 6, line 9: Generally, I think that this section describing the hypotheses and overall experimental goals is well-written. However, my concern here is how to introduce the second part of the study comparing N gas fluxes in small versus large holder systems in a more intuitive way. The current structure of this section makes the study on small versus large holder systems seem a bit disconnected from the first phase of the work. One possibility might be to introduce this study earlier on in the paragraph, close to the section where the authors pose their hypotheses (which implicitly refer to N availability and the HIP model), as this would then implicitly link-up to ideas about N control on N fluxes, e.g. (my suggestions in the underlined section below): "We covered four different land uses within two landscapes on highly weathered soils that mainly differed in soil texture (clay and loam Acrisols): forest, rubber trees inter spersed in secondary forest (hereafter called jungle rubber) as the reference land uses, and smallholder rubber and oil palm plantations as the converted land uses. In addi tion, we conducted a follow-on study comparing N gas fluxes across a gradient of N input that encompassed small holder plantations (lower N input rates) a large-scale oil palm plantations (higher N input rates) to try and evaluate the effect of N input rate on N gas fluxes..."

Answer: We greatly appreciate this referee's suggestion and we also see that we should bring out early on the part on the comparison of soil N2O fluxes between small-holder and large-scale oil palm plantations. We take this suggestion which is now incorporated in page 5, lines 13-19 of the revised manuscript.

2. Referee: Page 7, lines 12-17: In the comparison study between small holder versus large holder systems, were measurements from the small holder systems collected at the same time (i.e. were fluxes from the two types of oil plantations collected concomi-tantly)? If so, then this should be made clearer in this paragraph.

Answer: No, the measurements between the smallholder and large-scale plantations were not measured concomitantly, mainly because of logistical limitation. The measurement periods were also clearly stated in the original manuscript (pages 6-7, lines 24-6; page 7, lines 9-12; page 8, lines 22-24; page 9, lines 5-9), and now is clearly shown in Appendix Table A1 (see comment below). This was because our permission to work in the large-scale plantation was settled later than our agreement with the smallholders. However, this time difference in measurement period between these systems is accounted for in the statistical analysis - the linear mixed effect models include measurement period and replicate plot as random effects and only the oil palm plantation type as the fixed effect (Appendix A; page 47, lines 7-14).

3. Referee: Page 9, lines 7-17: It would be useful at the start of this paragraph to remind readers which land-uses were sampled in 2013, 2014 and 2015. Perhaps the authors could put together a table or something similar to represent this information?

Answer: We agree to these suggestions and put the measurement periods in an appendix table (now as Table A1 and the previous Table A1 now Table A2) for quick reference for the readers. We retain in the Methods (as referred to this page by the reviewer) all the description of these measurement periods and only give reference to Table A1 for summary.

4. Referee: Page 9, lines 18-20: Were the authors able to determine if N2O fluxes varied with distance from palms? Given the spatial structure in oil palm plantations, and the potential effects of roots and fertiliser application, it would be useful to know if the data could be corrected for spatial effects (if they exist) caused by proximity to palms.

Answer: The spatial structures of oil palms that are commonly seen in large-scale plantations are not consistent in smallholder plantations. The smallholder farmers also don't have regular spots for fertilizer applications, as we have explained in the manuscript based on our results. The deployment of the 4 permanently installed chamber bases per replicate plot is described in detail in section 2.1 "experimental design", page 6-7, lines 24-5. These chambers had random spatial locations in order to represent each replicate plot. These randomly placed chambers happened to be within 1.8 – 5-m distance to the palms and we conducted a Spearman's rank correlation test between N-oxide fluxes and distance to palms of the four replicate plots (sites) within each landscape. There were no significant correlations (P = 0.84–0.94). Thus, there was no basis for correction for any spatial effect as there exist no relationship with distance to the palms.

The best way to quantify any possible effects of fertilization is the way we described in our study, assuming that the random placement of chambers and the monthly sampling may have missed the fertilized spots and short-term effects of N application. Hence, we did the more intensive measurements following fertilization in the same smallholder plantations (using the same rate and application methods the smallholders claimed to employ) in order to quantify the contribution of fertilization, both in terms of space and duration, on our annual estimates.

5. Referee: Page 15, lines 16-25: I wonder if the large variation in the mean fluxes is driven by a high degree of within-plot spatial variability, which might linked to where fertiliser is applied, the distribution of palms, or surface residues (e.g. palm fronds or planted understory plants)? Is it possible to determine to what extent micro-scale variability, linked to spatial structure in the plantation, was causing variance in the measurements? This could help in interpreting the data, and understanding differences linked to management differences in small holder vs larger holder systems.

Answer: This question is related to our answer in #4 above. Fig. 1 shows the mean and SE from the 4 sites per land use on each sampling day. This variation, i.e. SE, on each sampling day reflected the variability among the 4 sites. The SE did not reflect within-plot variation because in the stat analysis (LME) the mean of the 4 chambers per plot (as subsamples nested within plot) on each sampling day is the value used in the LME analysis, which is conducted across all sampling days with land use as fixed effect and plot and sampling day as random effects (Appendix A, page 47, lines 2-3 and

6-7). Similarly, the statistically undetectable difference between the large-scale and smallholder oil palm plantations was also due to the large spatial variability among the 4 replicate plots and not by micro-scale variability within each plot. We have ascertained this in this large-scale plantation by statistical analysis since we have placed the 3 chambers/plot systematically to characterize any possible micro-scale variability within plots. In this large-scale plantation, we placed the 3 chambers/replicate plot such that the 1st chamber was on the fertilized band (at 0.8-1-m distance to the palm base) and the next 2 chambers were placed at a succeeding 2-m distance from each other. These 3 chamber locations, however, did not differ (P = 0.70) in soil N2O fluxes across the measurement period. This was due to the fact that the management practices were not consistently done as claimed by the plantation managers and smallholders. The field workers had sometimes broadcasted the fertilizers, sometimes also just applied in a band around the palm (page 8, lines 11-18), and sometimes piled the cut fronds in one row and sometimes in another row or sometimes not at all. The area and duration of effects of 1-2 times/yr fertilizer application in smallholders (which had 2-4 times lower application rates than in large-scale plantations) were small and only lasted for a few days (Figs. 2 & 3). We allocated full subsections of these effects in the Results and Discussion.

Even if we do a variance component analysis of the soil N2O fluxes, to partition the scales' contributions to the overall variance, this will not answer whether the within-plot variability is related to spatial structure of the management practices within plots. Variance component analysis will only quantify how much of the overall variability is accounted by within-plot variation. We think that the spatial structure of the management practices will only be detectable if there was a consistent management practices, e.g. when experimental plots are controlled by researchers. As we all know, smallholders as well as the large-scale plantations in reality do not have a uniform management practices in all years in terms of where fertilizers and residues are exactly placed, and hence we were unable to detect any statistically significant relationship of within-plot pattern of soil N2O fluxes with what is supposed-to-be the spatial structure management practices. That is however what was occurring in our actual field conditions. As we also did not find in the large-scale plantation correlation or differences in soil N2O fluxes between chamber locations and distance to palms, we also cannot relate within-plot variability to the spatial structure of this plantation. This is the main reason why we focused instead on our more frequent measurement following our own fertilization (mimicking farmers' claimed practice) to quantify the spatial and temporal contributions of fertilization on soil N2O fluxes (Fig. 2 & 3).

6. Referee: Page 16, lines 1-10: There is a potential confounding effect here due to the presence of roots which needs to be acknowledged. Granted, it is likely that the effect of fertiliser application will overwhelm the effect of roots in the immediate to short-term after fertilisation. However, it is worthwhile knowing whether or not the presence of roots ameliorates the effects of fertiliser (e.g. plant competition with nitrifiers/denitrifiers for inorganic N may reduce the relative gases loss of N in areas with high root densities). For example, do the authors have data on N gas fluxes from root-free and rhizosphere soil in the large holder systems to compare against? My thought here is that if the N application rate is higher in the large holder systems it may be possible to compare N fluxes from rhizosphere soil with different N application rates to evaluate the effect of N input rate on gas fluxes (i.e. making a like-for-like comparison).

Answer: From another study (conducted by another group in this collaborative project) that measured root distribution in the same smallholder oil palm plantations, there were no significant correlations between root mass distribution with distance to palms. This was attributed to the facts that these are mature plantations (12-16 yrs old, except one site that was 9 yrs old) and the weeding practices in smallholder plantations were not intensive (1-2 times per year only; Hassler et al., 2015) and hence the ground was almost always covered with undergrowth. It is impossible to see a root-free area. We don't think root can ameliorate the pulse effects of N fertilization on soil N2O fluxes (the total flux was also only 0.2-0.7% of the added N; page 17, lines 14-16), because we would have not seen a similar effect in tropical forest soils all covered with roots (e.g.

Koehler et al. 2009). In this latter study, soil N2O emissions clearly increase following N fertilization (at comparable application rate as we have in the present study) and went back to the background levels after about 6 weeks (during which the added N are already recycling within the soil N cycle). In the large-scale oil palm plantation PTPN VI (which was 12 yrs old), we don't have root data. Following Dr. Teh's suggestion of like-for-like comparisons, we conducted statistical analysis between the large-scale and smallholder plantations considering only the chambers on fertilized spots (the supposed-to-be spots which the smallholders and PNPT VI manager claimed where fertilizer was banded or broadcasted) and the sampling days within 6 weeks following fertilization; and a separate analysis for sampling days after six weeks of fertilization for chambers on locations which were not supposed-to-be fertilized. There was still no detectable significant difference between the large-scale and smallholder plantations (P = 0.50-0.67). Thus, the argument on confounding effect of roots was not convincing, at least from our dataset. The short-term effects of fertilizer application clearly showed the overwhelming effects (Figs. 2 & 3), although the emission percentage to amount of N added was actually only small; thus, presumably a large part of the added N must have been incorporated into the soil N cycle and eventually into the plant-soil cycling.

7. Referee: Page 16, lines 1-17: Regarding the use of locations a, b and c to refer to different distances to the palm; perhaps it may be possible to use identifiers that are a bit more descriptive, as this would make it easier for the readers to pick-up on the information quickly? e.g. 0.3 m = "inner root ball", 0.8 m = "outer root ball", 4-4.5 m = "inter-palm space" (or something similar)? Use of letters is a bit more abstract and (while clear) forces the reader to refer back to the tables or legends to remind themselves of the meaning of these abbreviations. Also – where trends are statistically significant, the authors could list the P-values from the multiple comparisons tests in parentheses to highlight where significant trends existed (I see that this has been done for the table, but would be useful for the reader if this was stated in the text, too).

Answer: We agree with the suggestion to use meaningful identifiers rather than a, b and c. To address this concern, we followed the suggestion of referee 1 and introduced the following clearer abbreviations: F1 = chamber location with incidental fertilization (0.3 m from the tree base), F2 = fertilized chamber location (0.8 m from the tree base), NF = non-fertilized chamber location (4Åň-4.5 m from the tree base) (page 10, lines 2-6 and Tables 2 & 4).

Furthermore, we indeed gave consistently the P values of all comparisons in the text and not just in the Table) (i.e. 1st and 3rd paragraphs in section 3.2 for comparisons among chambers).

8. Referee: Page 16, lines 18-22: Are these estimates derived from the trapezoidal extrapolations or some form of area-weighted upscaling?

Answer: The calculations for these estimates were explained in the Methods, pages 11-12, lines 18-4. From this equation, the total N-oxide emissions following fertilization (chambers F1 & F2) and the background fluxes (from the unfertilized chamber NF) are the trapezoidal calculations of the fluxes shown in Figs. 2 & 3 for each site/replicate plot. Since fertilizer-induced fluxes were limited in space and time, we also considered the fertilized area (multiplied by the tree density/ha) and the frequency of fertilizer application.

9. Referee: Page 18, section heading 3.3 Temporal controls of soil N-oxide fluxes: This section appears to discuss the relationship between environmental variables/drivers and N gas fluxes. Perhaps it may be more appropriate to re-name this section as "Role of abiotic variables in controlling N-oxide fluxes"? Or, if the authors may wish to more explicitly discuss how temporal variability in these environmental drivers contribute to fluctuations in N-oxide fluxes?

Answer: This page is still in the Result section – we mainly present (not discuss) the controlling factors of the temporal pattern of soil N-oxide fluxes. We keep this section heading, because we explicitly want to distinguish between temporal and spatial (section 3.4.) (and both sections considered abiotic factors) controls of N-oxide fluxes.

[Figure]

10. Referee: Page 18, section heading 3.4 Spatial controls of annual soil N2O fluxes: Similar to my above point (9), I do not feel that this heading properly describes what is discussed in the section. In this section, the authors discuss the relationship between N cycling processes rates and N-oxide fluxes, in order to evaluate the principal source of N oxides in these soils. They conclude that nitrification is probably the dominant driver of N-oxide fluxes because of the correlation between nitrification rates and gas fluxes. Perhaps the section could be retitled "Role of different N cycling processes in regulating N-oxide fluxes"? Also – I re-read the Methods and did not see the nitrification potential experiments described. Was this work done as part of another study or was this done as part of this work? In either case, this needs to be added to the Methods to make it clear that this work was done as the reference to nitrification (although interesting and relevant) came as a but of a surprise.

Answer: We have explicitly explained in the statistical analysis (page 14, lines 13-17) that for assessing the spatial controls of soil N2O fluxes, we used the annual flux per plot (and thus excluding the temporal variation) and conducted the correlation analysis with all the measured soil factors (physical and biochemical factors as well as the soil-N cycling processes) across the landscapes, encompassing soil conditions of the plots within the reference land uses and within the converted land uses. Thus, any significant relationships we observed suggested the range of conditions across plots and hence indicated the spatial controls. We also now added in the Methods (see answer to general comment #2 above) the descriptions of the sources of these soil controlling factors that were included in these correlation analysis for section 3.4. The control of gross nitrification was not interpreted as the main source of N2O fluxes in these soils but rather as an indicator of N availability in the soil; please see our answer to this similar comment in the 2nd general comment above.

11. Referee: Page 20, lines 9-22: Fluxes of NO from these systems, particularly oil palm, is extremely novel and of wider environmental significance, given the potential role of NO in tropospheric ozone formation, N deposition, and regional atmospheric oxidant (OH) balance. It would be useful in the discussion if the authors could bring into the discussion some of the findings from earlier atmospheric sampling campaigns by the OP3 consortium (Fowler et al., 2011, Hewitt et al., 2009), where elevated NOx concentrations were found in the troposphere near oil palm plantations? Hewitt et al. (2009) and Fowler et al. (2011) suggest that the implications of enhanced NO emission from oil palm could be potentially regionally significant, and the work here in Sumatra on ground-based NO fluxes would be an interesting counter-point to the atmospheric sampling work from Sabah.

Answer: We very much appreciated this suggestion and included this aspect in section 4.2 "Land-use change effects on soil N2O and NO fluxes from oil palm plantations" (page 23, lines 16-22).

———————————————

[Figure]

**Supplement:**

[revised manuscript text omitted]

| Oil palm (smallholder plantation) | 12.2 ± 6.1[a,A] | 0.7 ± 0.7[ab,A] | 1.1 ± 0.5
*0.1 ± 0.0* |
| Oil palm (large-scale plantation) | 42.3 ± 24.2[a,A] | - | 3.3 ± 1.7 |

**Table 2.** Mean (±SE, $n = 3$ oil palm trees) soil $N_2O$ and NO fluxes from three chamber locations during a fertilization in three (for $N_2O$) or one (for NO) smallholder oil palm plantation within each landscape, measured 6 to 11 times during 3–8.5 weeks following fertilization. Means followed by different letters indicate significant differences among chamber locations within each site (linear mixed-effect models with Fisher's LSD test at $P \leq 0.05$). Chamber F1, F2 and NF were placed at 0.3 m (with incidental fertilization), 0.8 m (fertilized area), and 4–4.5 m (non-fertilized area, serving as the reference chamber), respectively, from the stem base. 0.32 kg N tree$^{-1}$ was applied in the clay Acrisol and 0.26 kg N tree$^{-1}$ in the loam Acrisol in accordance to the smallholders' practices.

| Oil palm site | Chamber location | $N_2O$ fluxes ($\mu g\ N\ m^{-2}\ h^{-1}$) | NO fluxes ($\mu g\ N\ m^{-2}\ h^{-1}$) |
|---|---|---|---|
| clay Acrisol landscape | | | |
| 1 | F1 | $156.7 \pm 86.8^b$ | - |
| | F2 | $910.1 \pm 410.0^a$ | - |
| | NF | $6.9 \pm 3.3^c$ | - |
| 2 | F1 | $130.6 \pm 34.6^b$ | - |
| | F2 | $692.7 \pm 144.1^a$ | - |
| | NF | $9.9 \pm 3.0^c$ | - |
| 3 | F1 | $45.5 \pm 3.7^b$ | $4.7 \pm 1.7^b$ |
| | F2 | $1281.0 \pm 486.7^a$ | $535.3 \pm 194.5^a$ |
| | NF | $1.1 \pm 1.6^c$ | $1.5 \pm 1.5^b$ |
| Oil palm site | Chamber location | $N_2O$ fluxes ($\mu g\ N\ m^{-2}\ h^{-1}$) | NO fluxes ($\mu g\ N\ m^{-2}\ h^{-1}$) |
| loam Acrisol landscape | | | |

| | | | |
|---|---|---|---|
| 1 | F1 | $33.5 \pm 9.8^{b}$ | - |
| | F2 | $133.4 \pm 34.9^{a}$ | - |
| | NF | $11.8 \pm 6.1^{b}$ | - |
| | | | |
| 2 | F1 | $129.7 \pm 46.2^{a}$ | $46.2 \pm 19.6^{b}$ |
| | F2 | $205.3 \pm 24.2^{a}$ | $157.1 \pm 35.7^{a}$ |
| | NF | $7.9 \pm 4.8^{b}$ | $0.7 \pm 0.3^{b}$ |
| | | | |
| 3 | F1 | $5.2 \pm 1.0^{b}$ | - |
| | F2 | $104.5 \pm 81.9^{a}$ | - |
| | NF | $3.7 \pm 1.7^{b}$ | - |

**Table 3.** Pearson correlation coefficients between soil $N_2O$ flux ($n$ = 48; µg N m$^{-2}$ h$^{-1}$), soil

NO flux ($n$ = 16; µg N m$^{-2}$ h$^{-1}$), water-filled pore space (WFPS; %, top 0.05 m depth), soil temperature (°C, top 0.05 m depth) and extractable mineral N (mg N kg$^{-1}$, top 0.05 m depth)

across landscapes for the reference and converted land uses. Correlation was conducted using the means of the four replicate plots per land use on each of the 12 monthly measurements (for soil $N_2O$ fluxes) and four monthly-bimonthly measurements (for soil NO fluxes).

| Land-use type | Variable | WFPS | Soil temp. | $NH_4^+$ | $NO_3^-$ |
|---|---|---|---|---|---|
| Reference land uses (forest and jungle rubber) | Soil $N_2O$ flux | -0.21 | -0.09 | -0.23 | 0.38[c] |
| | Soil NO flux | -0.74[c] | -0.15 | -0.48[a] | 0.69[c] |
| Converted land uses (rubber and oil palm) | Soil $N_2O$ flux | 0.11 | 0.15 | 0.23 | 0.37[c] |
| | Soil NO flux | -0.05 | 0.09 | -0.05 | 0.23 |

[a]$P \leq 0.09$, [b]$P \leq 0.05$, [c]$P \leq 0.01$.

**Table 4.** Pearson correlation coefficients ($n$ = 6–11 measurements following fertilization)

between N-oxide fluxes ($\mu$g N m$^{-2}$ h$^{-1}$), water-filled pore space (WFPS; %, top 0.05m depth)

and extractable mineral N (mg N kg$^{-1}$, top 0.05 m depth), measured at different chamber locations (F1, F2 and NF were at 0.3 m (with incidental fertilization), 0.8 m (fertilized area)

and 4–4.5 m (non-fertilized area), respectively, from stem base). Correlation was conducted using the means of the three replicate trees per chamber location. 0.32 kg N tree$^{-1}$ was applied in the clay Acrisol and 0.26 kg N tree$^{-1}$ in the loam Acrisol in accordance to the smallholders'

practices.

| Oil palm plantation site | Chamber location | Variable | WFPS | $NH_4^+$ | $NO_3^-$ |
|---|---|---|---|---|---|
| clay Acrisol landscape | | | | | |
| 1 ($n$ = 6 measurements) | F1 | Soil $N_2O$ flux | 0.55 | 0.88[b] | 0.46 |
| | F2 | | 0.57 | -0.22 | -0.31 |
| | NF | | 0.37 | -0.64 | -0.44 |
| 2 ($n$ = 11 measurements) | F1 | Soil $N_2O$ flux | 0.11 | 0.93[c] | 0.95[c] |
| | F2 | | 0.08 | 0.05 | -0.06 |
| | NF | | 0.09 | -0.44 | -0.45 |
| 3 ($n$ = 10 measurements) | F1 | Soil $N_2O$ flux | -0.19 | 0.10 | 0.09 |
| | F2 | | 0.05 | 0.86[c] | 0.85[c] |
| | NF | | -0.32 | 0.06 | -0.44 |
| 3 ($n$ = 10 measurements) | F1 | Soil NO flux | -0.34 | 0.44 | 0.48 |
| | F2 | | -0.61[a] | 0.10 | -0.04 |
| | NF | | 0.59[a] | -0.14 | -0.13 |
| loam Acrisol landscape | | | | | |
| 1 ($n$ = 6 measurements) | F1 | Soil $N_2O$ flux | 0.96[c] | -0.18 | 0.03 |
| | F2 | | 0.78[a] | 0.61 | -0.40 |
| | NF | | -0.06 | -0.29 | <0.01 |

| | | | | | |
|---|---|---|---|---|---|
| 2 (n = 9 measurements) | F1 | Soil $N_2O$ flux | -0.55 | $0.71^b$ | -0.03 |
| | F2 | | 0.35 | -0.20 | $0.89^c$ |
| | NF | | 0.34 | <0.01 | -0.35 |
| 3 (n = 11 measurements) | F1 | Soil $N_2O$ flux | $-0.68^b$ | $0.67^b$ | $0.62^b$ |
| | F2 | | -0.27 | -0.2 | $0.57^a$ |
| | NF | | 0.36 | 0.19 | 0.06 |
| 2 (n = 9 measurements) | F1 | Soil NO flux | -0.07 | 0.18 | -0.27 |
| | F2 | | 0.07 | -0.11 | $0.96^c$ |
| | NF | | -0.16 | 0.12 | -0.23 |

$^aP \leq 0.09$, $^bP \leq 0.05$, $^cP \leq 0.01$.

[revised manuscript text omitted]

---

## Author Response (AR2)

**Author's responses**

We thank Dr. Paul Stoy for the time he invested to give his feedbacks to our manuscript. For clarity, we have copied his comments and questions and placed our answers below each comment/question.

*COMMENTS AND QUESTIONS:*

*1. Paul Stoy: How frequent are clay vs. loam Acrisol soils in Sumatra? Are both relatively common?*

Answer: Soil textural data are scarce, and so we cannot estimate the coverage of these two soil textural classes under this Acrisol soil group. However, this soil group (Acrisol soil; as the top level of soil classification in the FAO system), represents 50 % of land area in Sumatra (FAO et al., 2009). As Acrisol soil is dominant in area coverage (as the rest of 50 % is shared by many other soil groups), these two soil landscapes in our study area are indeed common in Sumatra.

*2. Paul Stoy: On page 9 (I emphasize that continuous line numbering is always strongly preferred) It is important to state how deep the chamber bases were installed. Too deep can damage belowground structures.*

Answer: We also see that it matters to specify how deep we inserted our chambers (~0.03 m) into the ground. To address this concern, we included this information on page 9, lines 13-14.

*3. Paul Stoy: On page 12, please use mathematical symbols for equations (the star means convolution and shouldn't be used as a surrogate for the multiplication symbol).*

Answer: We take this information and replaced the star symbol by the multiplication symbol ($\times$; page 12, lines 3-5).

*4. Paul Stoy: On line 14, don't use marginal significance here or anywhere else in the manuscript. Decide what is unlikely to happen by chance and stick with it.*

Answer: As we explain on page 14, lines 2-5, we considered also marginal significance since our experimental design covers the inherently high spatial variability of soil N-oxide fluxes. This marginal significant level was also accepted in our previous publication on soil $CO_2$ and $CH_4$ fluxes measured in the same study area, which was published in Biogeosciences (Hassler et al., 2015). In the present manuscript the marginal significance level serves especially to illuminate the differences in soil NO fluxes between rubber and jungle rubber in the loam Acrisol ($p = 0.07$). For these reasons we keep it in our manuscript.

*5. Paul Stoy: The paragraph on page 16 line 13 needs to be rewritten, it was difficult to follow.*

Answer: We re-structured this paragraph to improve clarity. We now state straightforwardly in the very beginning of the paragraph the main message and take out unnecessary information on fertilizer application times (page 16, lines 13-17).

*6. Paul Stoy:* What is SFB 990?

Answer: SFB stands for *Sonderforschungsbereich* or *Collaborative Research Center* (CRC) and this is financed by the German Research Foundation (DFG) within which we conducted our study. The description of CRC 990 is given in the acknowledgements. To improve clarity, we now replaced the German abbreviation with the English (CRC instead of SFB) (page 27, lines 19-20 and page 28, line 4). Also we referred to the acknowledgements on page 27, line 19, where the abbreviation is given the first time.

**References:**

FAO, IIASA, ISRIC, ISSCAS, JRC: Harmonized World Soil Database (version 1.1), available at: http://www.fao.org/soils-portal/soil-survey/soil-maps-and-databases/harmonized-world-soil-database-v12/en/ (last access: 05.04.2017), 2009.

[revised manuscript text omitted]

[a]$P \leq 0.09$, [b]$P \leq 0.05$, [c]$P \leq 0.01$.

[revised manuscript text omitted]